 

# Empathic pain evoked by sensory and emotional-communicative cues share common and process-specific neural representations

Feng Zhou[1,2]*, Jialin Li[1], Weihua Zhao[1], Lei Xu[1], Xiaoxiao Zheng[1], Meina Fu[1], Shuxia Yao[1], Keith M Kendrick[1], Tor D Wager[2], Benjamin Becker[1]*

[1]Clinical Hospital of Chengdu Brain Science Institute, MOE Key Laboratory for Neuroinformation, University of Electronic Science and Technology of China, Chengdu, China; [2]Department of Psychological and Brain Sciences, Dartmouth College, Hanover, United States

**Abstract** Pain empathy can be evoked by multiple cues, particularly observation of acute pain inflictions or facial expressions of pain. Previous studies suggest that these cues commonly activate the insula and anterior cingulate, yet vicarious pain encompasses pain-specific responses as well as unspecific processes (e.g. arousal) and overlapping activations are not sufficient to determine process-specific shared neural representations. We employed multivariate pattern analyses to fMRI data acquired during observation of noxious stimulation of body limbs (NS) and painful facial expressions (FE) and found spatially and functionally similar cross-modality (NS versus FE) whole-brain vicarious pain-predictive patterns. Further analyses consistently identified shared neural representations in the bilateral mid-insula. The vicarious pain patterns were not sensitive to respond to non-painful high-arousal negative stimuli but predicted self-experienced thermal pain. Finally, a domain-general vicarious pain pattern predictive of self-experienced pain but not arousal was developed. Our findings demonstrate shared pain-associated neural representations of vicarious pain.

*For correspondence:
zhou.feng@live.com (FZ);
ben_becker@gmx.de (BB)

Competing interests: The authors declare that no competing interests exist.

## Introduction

Pain empathy, the capacity to resonate with, relate to, and share others' pain, is an essential part of human experience. Among other functions, it motivates helping and cooperative behaviors and aids in learning to avoid harmful situations. Vicarious pain can be triggered by observing or imagining another individual's painful state and can be elicited by multiple types of social cues, particularly the observation of an inflicted physical injury or a facial expression of pain (*Decety and Ickes, 2009*; *Jauniaux et al., 2019*; *Vachon-Presseau et al., 2012*; *Yesudas and Lee, 2015*). While stimuli depicting the noxious stimulation of body limbs [i.e. observation of noxious stimulation (NS) induced vicarious pain (NS vicarious pain)] provide objective cues about the sensory component of the observed pain, the observation of facial expressions of pain [i.e. facial expressions induced vicarious pain (FE vicarious pain)] is considered more subjective and indirect as the pain experience of the expresser needs to be interpreted by the observer (*Hadjistavropoulos et al., 2011*; *Vachon-Presseau et al., 2012*). Functional magnetic resonance imaging (fMRI) studies employing corresponding pictorial stimuli have identified distinct and common neural substrates of pain empathy across vicarious pain induction procedures (*Jauniaux et al., 2019*). For example, *Vachon-Presseau et al., 2012* demonstrated that NS vicarious pain increased activity in core regions of the mirror neuron system, specifically inferior frontal and posterior regions engaged in coding sensory-somatic information

(*Rizzolatti and Craighero, 2004*) while the presentation of a facial expression of pain led to stronger increases in the medial prefrontal cortex and precuneus which have been associated with social cognitive processes such as mentalizing and theory of mind (*Amft et al., 2015*; *Gallo et al., 2018*; *Mitchell, 2009*; *Northoff et al., 2006*). Despite the different psychological domains engaged in the pain empathic responses induced by NS and FE both elicit vicarious pain experience (*Timmers et al., 2018*), encompassing pain-specific processes such as recognizing and understanding the painful state of the other person and affective sharing of pain but also non-specific processes that are shared between pain and other non-painful experiences such as arousal and negative affect (*Zaki et al., 2016*). In line with the shared underlying mental processes previous neuroimaging meta-analyses revealed that the observation of acute pain infliction and painful facial expressions commonly activate core empathy and nociceptive pain regions specifically the insular and cingulate cortices (*Jauniaux et al., 2019*; *Lamm et al., 2011*; *Timmers et al., 2018*). The overlapping activations have been suggested to reflect shared neural representations of vicarious pain (*Jauniaux et al., 2019*; *Lamm et al., 2011*; *Timmers et al., 2018*).

However, overlapping functional activations within these regions do not necessarily reflect shared underlying neural representations of a specific mental process (*Zaki et al., 2016*), given that (1) due to local spatial dependencies the main focus of traditional mass-univariate fMRI analytic approaches (i.e. conducting massive number of tests on brain voxels one at a time) is not on single-voxel activity, but on smoothed, regional differences in brain activity across multiple tasks or stimuli (*Haynes, 2015*) and (2) brain regions may contain multiple, distinct populations of neurons and averaging across those neuron populations yields nonspecific signals (*Haxby et al., 2014*; *Zaki et al., 2016*). For instance, electrophysiological and optogenetic studies have identified distinct neuronal populations in the anterior cingulate and insular cortices that activate during several functional domains, including pain- and empathy-related processes as well as attention, salience, social observation learning and reward expectancy (*Allman et al., 2011*; *Chen, 2018*; *Kvitsiani et al., 2013*; *Sakaguchi et al., 2018*; *Shidara and Richmond, 2002*; *Shura et al., 2014*; *Sikes and Vogt, 1992*). Studies employing mass-univariate fMRI analyses suggest that both regions are engaged by various experimental paradigms including not only experienced and observed pain, but also reward, arousal, salience and attention (*Cauda et al., 2012*; *Shackman et al., 2011*; *Uddin, 2015*; *Wager et al., 2016*; *Yarkoni et al., 2011*). Despite the overlapping fMRI activation in response to different experimental manipulations the underlying brain representations may be separable (*Corradi-Dell'Acqua et al., 2016*; *Krishnan et al., 2016*; *Woo et al., 2014*), emphasizing that more fine-grained analyses are required to determine process-specific shared or distinct neural representations (*Zaki et al., 2016*).

In an effort to overcome these limitations, recent studies have proposed several strategies to investigate the 'shared representation' question, including pharmacological (see e.g. *Rütgen et al., 2015*) and multivariate pattern analysis (MVPA) approaches. Compared to conventional analytic approaches, MVPA can be effective in extracting information at much finer spatial scales (e.g. below the intrinsic resolution determined by the voxel size by pooling together weak feature-selective signals in each voxel; *Kamitani and Tong, 2005*; *Woo et al., 2017*) and represents a more suitable approach to support or reject claims about neural mechanisms that are shared between mental processes (*Chikazoe et al., 2014*; *Peelen and Downing, 2007*; *Zaki et al., 2016*). In support of this view, using MVPA approaches researchers have demonstrated shared neural representations across mental processes (including self-experienced and observed pain) in both humans and animals (*Carrillo et al., 2019*; *Corradi-Dell'Acqua et al., 2011*; *Corradi-Dell'Acqua et al., 2016*). Moreover, a growing number of recent studies have demonstrated functional independence of overlapping univariate activation in these brain regions using MVPA (*Krishnan et al., 2016*; *Peelen et al., 2006*; *Woo et al., 2014*), including separable neural representations of physical and social rejection pain within the dorsal anterior cingulate cortex (*Woo et al., 2014*) and of modality-specific aversive experience in the anterior insular cortex (*Krishnan et al., 2016*).

Nevertheless, shared multivariate patterns do not necessarily imply process-specific common neural representations per se given that the shared neural representations could simply reflect common demands on basal processing domains such as attention or arousal (*Corradi-Dell'Acqua et al., 2011*; *Corradi-Dell'Acqua et al., 2016*; *Krishnan et al., 2016*). For instance, *Corradi-Dell'Acqua et al., 2016* found shared neural patterns between vicarious and self-experienced pain in the left anterior insula and further demonstrated that the common local patterns were not specific

to pain-related processing, but also represented disgust and unfairness suggesting modality-unspecific processing of aversive and arousing experiences.

This leads to the questions of (1) whether or not NS and FE-induced vicarious pain share pain-associated common neural representations, and further (2) whether a general (i.e. across NS and FE vicarious pain modalities) neural signature of vicarious pain, which is specific to the pain empathic response rather than capturing unspecific processes such as negative emotional experience or arousal, can be determined. More specifically, we examined the following three questions in this study: (i) whether NS and FE-induced vicarious pain-predictive signatures share spatially (correlation and distribution) and functionally (predictions of cross-modality vicarious pain versus corresponding non-painful control stimuli) similar neural representations, (ii) whether a general and specific vicarious pain-predictive neural signature, which should (1) generalize across different vicarious pain stimuli and (2) not be sensitive to predict unspecific negative affect or arousal induced by non-painful negative stimuli, and (3) be 'activated' by the direct experience of somatic pain as reflected by an accurate prediction of self-experienced somatic pain, can be determined.

To this end, we employed MVPA to fMRI data from an experiment during which participants were presented with stimuli depicting the infliction of noxious stimulation of body limbs (NS vicarious pain) or painful facial expressions (FE vicarious pain) as well as corresponding non-painful control stimuli (*Figure 1A*). Given that relative to their control stimuli both sets of painful stimuli were perceived as more painful in terms of recognized and shared pain as well as more arousing and negative

## A Examples of vicarious pain-evoking and control images

**NS vicarious pain**    **NS control**    **FE vicarious pain**    **FE control**

## B Behavioural ratings of the stimuli

**Figure 1.** Examples and behavioral ratings of the experimental stimuli. (**A**) Examples of stimuli for NS and FE vicarious pain as well as corresponding non-painful control stimuli. Of note, examples of the facial expressions (FE) were not included in the original stimulus set and written consent was obtained from the two volunteers. (**B**) Behavioral ratings of the stimuli from an independent sample (n = 38). Error bars represent standard errors of the mean. 'Other's pain' indicates 'how much pain do you think the person in the photo is feeling', 'self-pain' indicates 'how much pain do you experience when watching the picture'. All ratings were assessed by nine-point Likert scales ranging from '1 = not painful at all or very negative or very low arousing' to '9 = extremely painful or very positive or very high arousing'. NS vicarious pain, observation of noxious stimulation of body limbs induced vicarious pain; FE vicarious pain, observation of facial expressions of pain induced vicarious pain; NS control stimuli depict body limbs in similar but innocuous situations, FE control stimuli show neutral facial expressions.

(details see Results and *Figure 1B*), we additionally asked participants to undergo an emotion processing paradigm with non-painful high-arousal negative stimuli and low-arousal neutral stimuli from the International Affective Picture System (IAPS) (details see Materials and methods) to further test the specificity of the shared neural representations with respect to the vicarious experience of pain rather than emotional arousal or negative affect. To determine the association of the vicarious pain signature with direct pain experience, we included an independent fMRI dataset that collected ratings of self-experienced pain during thermal pain induction (details are provided in Materials and methods and in *Wager et al., 2013*; *Woo et al., 2015*).

Given that small sample sizes may lead to a large cross-validation error which is the discrepancy between the prediction accuracy measured by cross-validation and the expected accuracy on new data (*Varoquaux, 2018*) and fMRI-based inferences on regions that are most predictive substantially benefit from larger samples (*Chang et al., 2015*) we included a comparably large sample of n = 238 individuals (details see Materials and methods and *Li et al., 2018*; *Xu et al., 2020a*).

## Results

### Evaluation of the stimuli

To match the instructions between the vicarious pain and negative emotion fMRI paradigms an implicit instruction was provided (attentively view the pictorial stimuli) (details see Materials and methods). Affective ratings of the stimuli in an independent sample confirmed that both sets of painful stimuli were rated as considerably more painful compared to their respective control stimuli, both in terms of recognizing and sharing pain, and additionally were rated as more arousing and negative. Specifically, both categories of painful stimuli elicited a substantial level of pain intensity perceived for the person displayed as well as in the observer. The NS vicarious pain stimuli were rated considerably higher on both dimensions (mean ± standard error (SE) pain intensity displayed = 6.73 ± 0.27; mean ± SE pain intensity self-experienced = 6.14 ± 0.36) as compared to the corresponding NS control stimuli (mean ± SE pain intensity displayed = 1.37 ± 0.11; mean ± SE pain intensity self-experienced = 1.54 ± 0.14; $t_{37}$ = 18.11, p<0.001; $t_{37}$ = 12.71, p<0.001, respectively). Similarly, the FE vicarious pain stimuli were also rated substantially higher on both pain-related dimensions (mean ± SE pain intensity displayed = 6.20 ± 0.25; mean ± SE pain intensity self-experienced = 5.05 ± 0.31) as compared to the corresponding FE control stimuli (mean ± SE pain intensity displayed = 1.78 ± 0.16; mean ± SE pain intensity self-experienced = 2.20 ± 0.18; $t_{37}$ = 14.58, p<0.001; $t_{37}$ = 9.00, p<0.001, respectively) (*Figure 1B*). Moreover, both categories of painful stimuli were rated as considerably more negative and induced stronger arousal in the participants as compared to their respective control stimuli (NS vicarious pain stimuli: mean ± SE valence=3.14 ± 0.22; mean ± SE arousal=5.81 ± 0.32; NS control stimuli: mean ± SE valence=5.12 ± 0.15; mean ± SE valence=2.68 ± 0.25; $t_{37}$ = −7.99, p<0.001; $t_{37}$ = 9.02, p<0.001, respectively; FE vicarious pain stimuli: mean ± SE valence=3.57 ± 0.21; mean ± SE arousal=5.03 ± 0.29; FE control stimuli: mean ± SE valence=4.83 ± 0.12; mean ± SE valence=3.34 ± 0.24; $t_{37}$ = −5.24, p<0.001; $t_{37}$ = 6.50, p<0.001, respectively) (*Figure 1B*).

Likewise, the non-painful negative IAPS pictures were rated as considerably more arousing and negative as compared to the corresponding neutral stimuli. Specifically, negative stimuli elicited substantial negative affect and arousal on numerical rating scales (mean ± SE valence=2.41 ± 0.16; mean ± SE arousal=6.34 ± 0.22) compared with neutral stimuli (mean ± SE valence=5.35 ± 0.08; mean ± SE arousal=3.22 ± 0.25; $t_{36}$ = −16.09, p<0.001; $t_{36}$ = 12.65, p<0.001, respectively).

Post-fmri ratings further confirmed that vicarious pain stimuli elicited higher recognizing pain and arousal as compared to the control stimuli (*Supplementary file 1*). Of note, although we found that the two vicarious pain-evoking stimulus sets were not fully matched in terms of vicarious pain intensity, arousal and valence, the differences between the stimulus sets may only have a small effect on our findings given that (1) this study focused on common rather than different empathic pain responses elicited by the two stimulus sets and (2) both categories of vicarious pain stimuli elicited substantial levels of pain empathy.

## Univariate approach - shared activations for NS and FE vicarious pain

To test whether NS and FE vicarious pain share similar activation patterns as determined by traditional mass-univariate analyses, we performed a permutation-based correlation analysis to compare the spatial similarity between the unthresholded group-level NS vicarious pain activation (NS vicarious pain >NS control) and the FE vicarious pain activation (FE vicarious pain >FE control). We found that activation in response to NS vicarious pain was spatially correlated with that to FE vicarious pain (r = 0.171, p<0.1 based on permutation tests) (*Figure 2A*). Moreover, after multiple comparisons correction (FDR corrected, *q* < 0.05, two-tailed) (*Figure 2B and C*), distributed regions of overlapping activation were identified, including a network exhibiting increased activation during both modalities encompassing the bilateral anterior and mid-insula, dorsomedial prefrontal cortex, inferior parietal lobule, middle frontal gyrus and middle temporal gyrus, as well as a network of decreased activation, including the rostral and ventral anterior cingulate cortices, ventromedial and orbitofrontal cortices, and lingual and parahippocampal gyri (*Figure 2D*).

## Multivariate approach – modality general vicarious pain-predictive patterns

Previous studies have suggested that pain and negative emotional processes are distributed across brain regions (*Chang et al., 2015*; *Krishnan et al., 2016*; *Wager et al., 2013*) and that compared to whole-brain predictive models local regions explain considerably less variance in predicting these processes (*Kragel et al., 2018*; *Woo et al., 2017*). In an initial step, we therefore developed novel whole-brain patterns to decode NS and FE vicarious pain separately. The NS vicarious pain-predictive pattern yielded an average classification accuracy of 88 ± 1.5% SE, p<0.001, d = 2.13; d indicates effect size in terms of Cohen's d (accuracy = 96 ± 1.2% SE, p<0.001, d = 2.17 based on a two-alternative forced-choice test) and the FE vicarious pain-predictive pattern discriminated FE vicarious pain versus FE control with 80 ± 1.8% SE accuracy, p<0.001, d = 1.64 (accuracy = 88 ± 2.1% SE, p<0.001, d = 1.57 based on a two-alternative forced-choice test) with a 10-fold cross-validation procedure which was repeated 10 times, yielding 10 random partitions of the original sample.

Next, permutation-based correlation analysis was employed to determine the similarity between the whole-brain patterns of NS and FE vicarious pain which confirmed that the modality-specific patterns were spatially correlated (r = 0.170, p<0.001 based on permutation tests) (*Figure 3A*). To further qualitatively determine shared but also distinct vicarious pain signatures we analyzed the spatial covariation between the unthresholded weight maps for NS and FE vicarious pain. To this end, we plotted the joint distribution of normalized (z-transformed) voxel weights of the FE vicarious pain-predictive pattern on the x-axis and the NS vicarious pain-predictive patterns on the y-axis in *Figure 3B* (for similar approach see *Koban et al., 2019*; *Yu et al., 2019*). Briefly, pattern weights in any given voxel are expressed as positive, negative or near-zero values for each of the vicarious pain-predictive patterns, which allows to divide voxels into eight equally sized Octants depending on the relative weights in each pattern. For visual presentation, the Octants were color-coded with different colors indicating either voxels of shared positive or shared negative weight (Octants 2 and 6, respectively), selectively positive weights for NS (Octant 1) and FE (Octant 3) vicarious pain-predictive patterns, selectively negative weights for either NS (Octant 5) or FE (Octant 7) vicarious pain-predictive patterns, or opposite weights in the two decoders such that voxels in Octants 4 and 8 express positive and negative weights for the FE vicarious pain-predictive pattern but negative and positive weights for NS vicarious pain-predictive pattern, respectively. Furthermore, to provide an overall measure for voxels in each Octant, we computed the sum of squared distances (SSD) from the origin, which accounts for both, absolute numbers of voxels in each Octant and their (squared) distance from the origin. This analysis of the spatial coactivation of NS and FE vicarious pain-predictive patterns revealed peak SSDs in Octants 2 and 6 as compared to other Octants, suggesting that a considerable number of voxels express positive or negative weights for both vicarious pain-predictive patterns. Overall, this analysis provides further supports largely shared, but also non-shared, neural representations for NS and FE vicarious pain. In support of this, between-modality classifications showed that the NS vicarious pain-predictive pattern could reliably discriminate FE vicarious pain versus FE control with 69% accuracy (±3.0% SE, p<0.001, d = 0.65) and that the FE vicarious pain-predictive pattern could discriminate NS vicarious pain versus NS control with 78% accuracy (±2.7% SE, p<0.001, d = 1.00) based on two-alternative forced-choice tests with a repeated 10-fold

## A  Activation similarity between NS and FE vicarious pain

$r = 0.171^{\dagger}$

## B  NS vicarious pain > control (FDR $q$ < 0.05)

## C  FE vicarious pain > control (FDR $q$ < 0.05)

## D  Overlapping regions from the GLM analyses

**Figure 2.** Results from the conventional univariate analyses. (**A**) The NS vicarious pain activation pattern was spatially correlated with the FE vicarious pain pattern. (**B**) Results from the conventional univariate analysis comparing NS vicarious pain with the NS control stimuli thresholded at FDR $q$ < 0.05 (two-tailed). (**C**) Results from the univariate analysis comparing FE vicarious pain with the FE control stimuli thresholded at FDR $q$ < 0.05 (two-tailed). (**D**) Overlapping activation between NE and FE vicarious pain as determined by the conventional univariate approach. $^{\dagger}$p<0.1. NS vicarious pain, observation of noxious stimulation of body limbs induced vicarious pain; FE vicarious pain, observation of facial expressions of pain induced pain; NS control stimuli depict limbs in similar but innocuous situations, FE control stimuli show neutral facial expressions.

cross-validation procedure (*Figure 3C*). Taken together, our results confirmed shared neural representations between the different vicarious pain modalities at the whole-brain level, yet the reduced between-modality prediction effect sizes as compared to within-modality prediction effect sizes (<50%) additionally suggest distinguishable neural representations. Results remained significant after correcting for multiple comparisons using Bonferroni correction.

## A  Spatial similarity between whole-brain NS and FE vicarious pain patterns

$r = 0.170***$

## B  Joint distribution of normalized whole-brain weights of NS and FE vicarious pain patterns

## C  Cross-validated accuracy in two-choice classification tests (whole-brain patterns)

**Figure 3.** Results from the whole-brain multivariate pattern analyses. (**A**) The NS vicarious pain-predictive pattern was spatially correlated with the FE vicarious pain-predictive pattern. (**B**) Scatter plot displaying normalized voxel weights for NS (y-axis) and FE (x-axis) vicarious pain-predictive patterns. Bars on the right represent the sum of squared distances from the origin (0,0) for each Octant. Different colors are assigned to the eight Octants that reflect voxels of shared positive or shared negative weights (Octants 2 and 6, respectively), selectively positive weights for the NS (Octant 1) or for FE (Octant 3) vicarious pain patterns, selectively negative weights for the NS (Octant 5) or FE (Octant 7) vicarious pain patterns, and voxels with opposite weights for the two neural signatures (Octants 4 and 8). The numbers on the top of each bar indicate the voxel counts for each Octant. (**C**) Cross-

*Figure 3 continued on next page*

*Figure 3 continued*

validation accuracy as determined by two-alternative forced-choice classification tests based on the whole-brain patterns. The results demonstrated significant within- and between- modality classifications for both NS and FE vicarious pain-predictive patterns. The dashed line indicates the chance level (50%), and the error bars represent the standard error of the mean across subjects. ***p < 0.001. SSD, sum of squared distances. Error bar indicates standard error. NS vicarious pain, observation of noxious stimulation of body limbs induced vicarious pain; FE vicarious pain, observation of facial expressions of pain induced pain.

## Shared local representations for NS and FE vicarious pain

To identify brain regions which made reliable contributions to both whole-brain NS and FE vicarious pain-predictive patterns, we thresholded the corresponding neural patterns at FDR $q < 0.05$ (two-tailed, bootstrap tests with 10,000 iterations) separately and found overlapping regions in the bilateral mid-insula, left putamen and left inferior parietal lobule (*Figure 4A, B and C*), emphasizing the importance of these regions for encoding both NS and FE vicarious pain. Moreover, we employed a searchlight-based approach to locate regions which could predict both within-modality vicarious pain (e.g. NS vicarious pain-predictive patterns to predict NS vicarious pain versus NS control) as well as between-modality vicarious pain (e.g. NS vicarious pain-predictive patterns to predict FE vicarious pain versus FE control) using a cross-validation procedure. We found that a bilateral network encompassing the insula, striatum as well as the ventromedial prefrontal cortex (see *Figure 4D*, $q < 0.05$, FDR corrected, two-tailed) demonstrated significant within-modality cross-validation and between-modality cross-prediction accuracies between NS and FE vicarious pain, implying shared representation at the local pattern level. We additionally re-ran searchlight analyses with two different searchlight sizes (4-mm- and 10-mm-radius spheres) and found that the overlapping vicarious pain networks remained robust across different searchlight sizes (details see *Figure 4—figure supplement 1*).

## Shared representations in the mid-insula

Across the analyses, we observed overlapping activation and shared representations in the mid-insula (see also *Figure 4—figure supplement 1* for convergent findings across searchlight sizes). Accumulating evidence suggest a critical role of the mid-insula in pain-related processes, including self-experienced as well as vicarious pain. In line with functional anatomical studies suggesting that the mid-insula receives nociceptive information from thalamic nuclei (*Craig et al., 1994*; *Craig et al., 2000*) intracerebral electrical stimulation of the mid-insula evokes pain sensations (*Afif et al., 2010*) and previous MVPA studies demonstrated distinct neural representations between pain and non-pain negative stimuli in the (right) mid-insula yet shared representations across self-experienced and vicarious pain (*Corradi-Dell'Acqua et al., 2011*), while a recent meta-analysis of conventional fMRI empathy studies reported that vicarious pain uniquely activates the bilateral mid-insula and MCC as compared to empathy for non-pain negative affective states (*Timmers et al., 2018*). Based on the specific role of the mid-insula in pain-related processes, we further explored whether the mid-insula shared neural representations of NS and FE could be sufficient to predict vicarious pain. The mid-insula was defined based on the Human Connectome Project (HCP) multi-modal parcellation atlas (*Glasser et al., 2016*) (encompassing Pol2, FOP2, FOP3 and MI and available from the Cognitive and Affective Neuroscience Laboratory Github repository at https://github.com/canlab/Neuroimaging_Pattern_Masks; *Figure 5—figure supplement 1* displays the mid-insula mask). We found that NS vicarious pain activation in the insula was strongly positively correlated with FE vicarious pain activation (r = 0.737, p=0.006 based on permutation tests) and consistent with this, that the NS vicarious pain-predictive and FE vicarious pain-predictive pattern weights within the mid-insula were also strongly positively correlated (r = 0.538, p<0.001 based on permutation tests) (*Figure 5A and B*). Moreover, plotting the amount of shared positive, shared negative, and unique positive and negative voxel weights (z-scored) within the mid-insula for NS and FE vicarious pain-predictive patterns indicated that most voxels in the mid-insula exhibited shared positive weights (Octant 2) or negative weights (Octant 6), whereas only few voxels exhibited opposite weights directions (Octants 4 and 8) (*Figure 5C*). Consistent with the voxel-wise weight distribution two-alternative forced-choice tests revealed that the mid-insula partial NS vicarious pain-predictive pattern classified above chance for both, NS vicarious pain versus NS control (71 ± 2.9% SE, p<0.001, d = 0.72; within-modality) and FE

**A  NS vicarious pain-predictive pattern (FDR *q* < 0.05)**

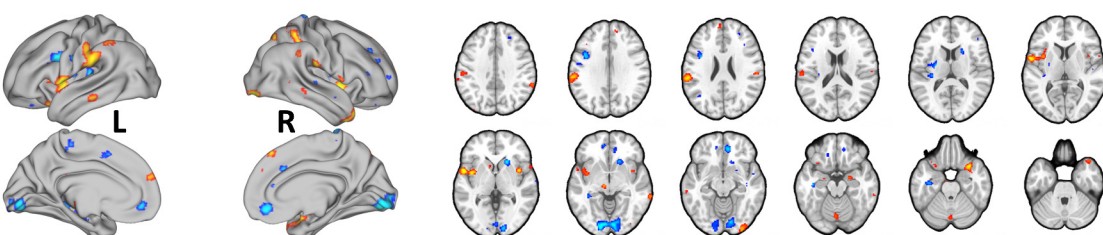

**B  FE vicarious pain-predictive pattern (FDR *q* < 0.05)**

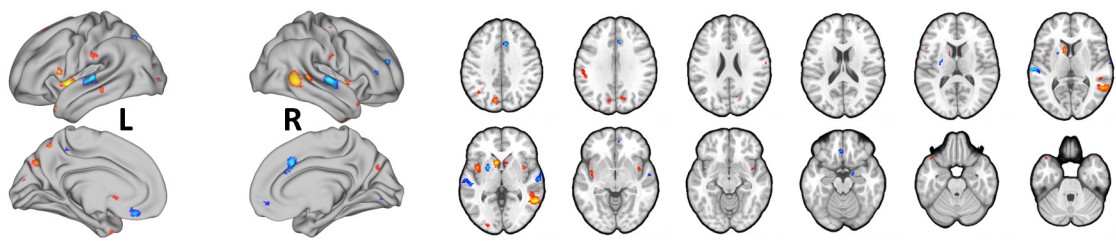

**C  Overlapping regions from whole-brain multivariate pattern analyses (FDR *q* < 0.05)**

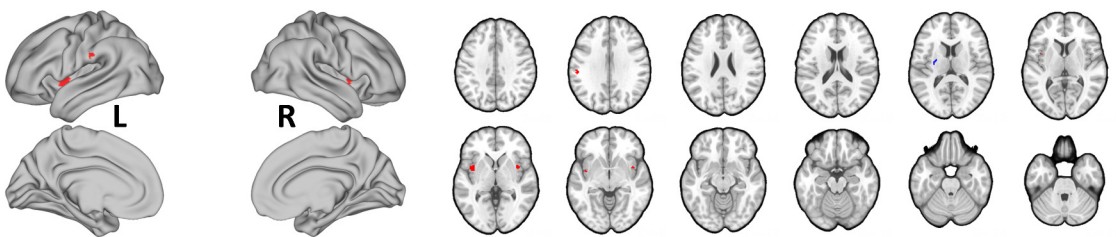

**D  Overlapping regions from the searchlight-based multivariate pattern analyses (FDR *q* < 0.05)**

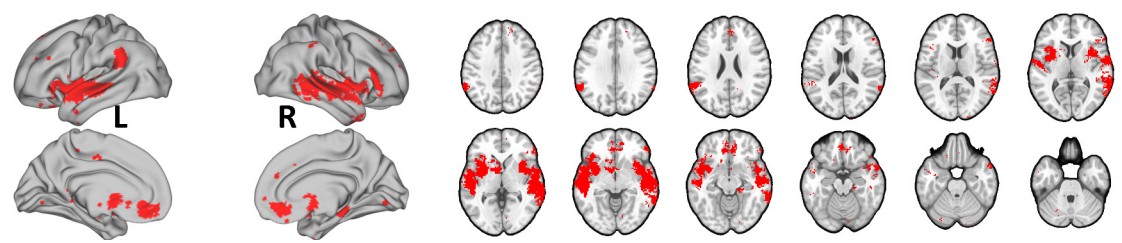

**Figure 4.** Brain regions that made reliable contributions to decoding vicarious pain. NS (**A**) and FE (**B**) vicarious pain-predictive patterns and (**C**) overlapping reliable predictive voxels (bootstrap thresholded at FDR *q* < 0.05, two-tailed). (**D**) Brain regions exhibiting significant within-modality cross-validation and between-modality cross-prediction accuracies between NS and FE vicarious pain (thresholded at FDR *q* < 0.05, two-tailed). NS vicarious pain, observation of noxious stimulation of body limbs induced vicarious pain; FE vicarious pain, observation of facial expressions of pain induced pain.
*Figure 4 continued on next page*

*Figure 4 continued*

The online version of this article includes the following figure supplement(s) for figure 4:

**Figure supplement 1.** Searchlight analyses with different searchlight sizes.

vicarious pain versus FE control (61 ± 3.2% SE, p<0.001, d = 0.36; between-modality prediction) in out-of-sample participants through a repeated 10-fold cross-validation procedure. In line with this, the mid-insula partial FE vicarious pain-predictive pattern discriminated NS vicarious pain versus NS control with 65% accuracy (±3.1% SE, p<0.001, d = 0.58; between-modality) and FE vicarious pain versus FE control with 60% accuracy (±3.2% SE, p=0.004, d = 0.27; within-modality) (*Figure 5D*). Together, these findings converge on common representations of vicarious pain in the mid-insula across univariate and multivariate patterns for NS and FE vicarious pain. However, although statistically significant, thus reflecting that the mid-insula plays important roles in encoding NS and FE vicarious pain and that the neural representations of NS and FE vicarious pain in this region are similar, the much lower effect sizes (as compared with the whole-brain predictions) indicate that the mid-insula is not sufficient to capture vicarious pain processing alone. Results remained significant after correcting for multiple comparisons using Bonferroni correction.

## Shared vicarious pain representations are not sensitive to arousal or negative affect

One key question is whether the developed vicarious pain-predictive patterns are specific to the vicarious sharing of pain or are rather generally sensitive to emotional arousal or negative affect. To test the functional specificity, whole-brain patterns were separately employed to discriminate processing of high-arousal non-painful negative from low-arousal neutral stimuli from the IAPS database with two-alternative forced-choice tests through a repeated 10-fold cross-validation procedure. This approach revealed statistically significant yet comparably low accuracies and small effect sizes (NS vicarious pain-predictive pattern: 58 ± 3.2%, p=0.024, d = 0.34; FE vicarious pain-predictive pattern: 61 ± 3.2% SE, p=0.001, d = 0.42). In contrast, testing whether shared representations in the mid-insula could discriminate negative versus neutral stimuli revealed that neither of the insula partial patterns could classify negative stimuli above chance level (NS: 56 ± 3.2% SE, p=0.079, d = 0.11; FE: 56 ± 3.2% SE, p=0.111, d = 0.09), suggesting a pain-specific representation in this region.

In addition, using the emotional processing data we developed a negative emotion-predictive pattern which could accurately classify non-painful negative vs. neutral stimuli (accuracy = 86 ± 1.6% SE, p<0.001, d = 2.07 using a repeated 10-fold cross-validation procedure). The negative emotion-predictive pattern could significant discriminate NS vicarious pain versus its control (cross-validated accuracy = 70 ± 3.0% SE, p<0.001, d = 0.88) and FE vicarious pain versus its control (cross-validated accuracy = 61 ± 3.2% SE, p<0.001, d = 0.28). However, accuracy and effect size are lower as compared to FE vicarious pain pattern's prediction of NS vicarious pain (cross-validated accuracy = 78 ± 2.7% SE, p<0.001, d = 1.00) and vice versa (cross-validated accuracy = 69 ± 3.0% SE, p<0.001, d = 0.65) and the mid-insula negative-predictive pattern did not predict vicarious pain (accuracies = 40 ± 3.2% SE, 48 ± 3.2% SE for NS and FE vicarious pain, respectively). Moreover, in contrast to the pain-predictive patterns (see below for details) neither the whole-brain nor the mid-insula negative-predictive pattern could predict thermal pain intensity (whole-brain, $r_{196} = 0.101$, p=0.157; mid-insula, $r_{196} = -0.319$), which additionally emphasizes the functional specificity of the pain-predictive pattern in the domain of pain-related processing. Together these findings suggest that negative emotional processing might share some neural representations with vicarious pain, but that the whole-brain and mid-insula vicarious pain representations are more specific to the pain-related information. Results remained stable after correcting for multiple comparisons using Bonferroni correction.

## A vicarious pain-predictive pattern that predicts both NS and FE vicarious pain

Given that the NS and FE vicarious pain-predictive patterns shared similar whole-brain as well as local neural representations, we developed a general vicarious pain pattern which yielded a classification accuracy of 82 ± 1.2% SE, p<0.001, d = 1.77 (accuracy = 91 ± 1.3% SE, p<0.001, d = 1.74 based on a two-alternative forced-choice test) in discriminating vicarious pain versus non-painful

**A Spatial similarity between mid-insula NS and FE vicarious pain activations**

L    R

*r* = 0.737**

L    R

**B  Spatial similarity between mid-insula NS and FE vicarious pain patterns**

L    R

*r* = 0.538***

L    R

**C  Joint distribution of normalized mid-insula weights of NS and FE vicarious pain patterns**

**D  Cross-validated accuracy in two-choice classification tests (mid-insula patterns)**

**Figure 5.** Results of the mid-insula focused analyses. (**A**) Mid-insula activation to NS vicarious pain was highly similar to activation to FE vicarious pain. (**B**) NS vicarious pain-predictive pattern in the mid-insula was spatially similar to the FE vicarious pain-predictive pattern. (**C**) Examining voxel-level similarity in bilateral mid-insula revealed that that the majority of mid-insula voxels exhibited shared positive or negative weights (Octants 2 and 6, respectively). Selective weights are depicted as: selective positive weights for NS (Octant 1) and for FE (Octant 3) vicarious pain patterns, selective

*Figure 5 continued on next page*

*Figure 5 continued*

negative weights for NS (Octant 5) and for FE (Octant 7) vicarious pain patterns. Voxels with opposite weights for the two signatures are depicted in Octants 4 and 8. (**D**) Cross-validation accuracy from the two-choice classification tests with mid-insula partial patterns. The results demonstrated significant within- and between-modality classifications for both NS and FE vicarious pain-predictive patterns. The dashed line indicates the chance level (50%), and error bars represent standard error of the mean across subjects. $^{**}p < 0.01$; $^{***}p < 0.001$. SSD, sum of squared distances. Error bar indicates standard error.

The online version of this article includes the following figure supplement(s) for figure 5:

**Figure supplement 1.** The mid-insula mask used in the current study.

control. More specifically, the pattern could accurately predict both NS vicarious pain from the NS control (95 ± 1.4% accuracy, p<0.001, d = 2.10) and FE vicarious pain from the FE control (87 ± 2.1% accuracy, p<0.001, d = 1.45), but performed considerably worse classifying non-painful negative versus neutral stimuli (59 ± 2.1% accuracy, p=0.01, d = 0.30), in forced-choice classifications. In line with the spatially overlapping modality-specific vicarious pain patterns the general vicarious pain pattern was highly similar with both, the NS vicarious pain pattern (r = 0.587, permutated p<0.001 based on permutation tests) and FE vicarious pain pattern (r = 0.702, p<0.001 based on permutation tests). To functionally characterize the general vicarious pain-predictive pattern the Neurosynth decoder function was used to assess its similarity to the reverse inference meta-analysis maps generated for the entire set of terms included in the Neurosynth dataset. The most relevant features were 'painful' and 'pain' for the top 50 terms (excluding anatomical terms) ranked by the correlation strengths between the vicarious pain pattern and the meta-analytic maps (see word cloud, size of the font scaled by correlation strength, *Figure 6A*). After thresholding and correction for multiple comparisons (bootstrapping 10,000 samples, FDR q < 0.05, two-tailed), the general vicarious pain-predictive pattern revealed a distributed network engaged in vicarious pain processing encompassing the bilateral mid-insula, inferior parietal lobule and ventromedial prefrontal cortex (*Figure 6B*), further emphasizing the importance of these regions for encoding vicarious pain. All conclusions remained stable after controlling for multiple comparisons using Bonferroni correction.

## Association of the vicarious pain-predictive pattern with self-experienced somatic pain

To test the associations between the vicarious pain representation with directly experienced pain, we applied the whole-brain general vicarious pain-predictive pattern to self-experienced thermal pain data using dot-product of vectorized activation maps with the pattern classifier weights. We found that the general vicarious pain-predictive pattern expressions were highly correlated with both overall objective temperature levels ($r_{196}$ = 0.538, p<0.001) and subjective pain ratings ($r_{196}$ = 0.507, p<0.001). Moreover, the general pain-predictive pattern discriminated high thermal pain versus low thermal pain with a 94% accuracy (±4.2% SE, p<0.001, d = 2.00), high thermal pain versus medium thermal pain with a 91% accuracy (±5.0% SE, p<0.001, d = 1.56) and medium thermal pain versus low thermal pain with an 82% accuracy (±6.7% SE, p=0.001, d = 1.20) using two-alternative forced-choice tests (*Figure 7A*).

When prediction focused on the mid-insula the general vicarious pain-predictive local pattern could discriminate high thermal pain versus low thermal pain (accuracy = 88 ± 5.7% SE, p<0.001, d = 1.56), high thermal pain versus medium thermal pain (accuracy = 88 ± 5.7% SE, p<0.001, d = 1.23) and medium thermal pain versus low thermal pain (accuracy = 82 ± 6.7% SE, p<0.001, d = 1.49) above chance levels (*Figure 7B*). In addition, the mid-insula partial pattern expressions (i.e. focusing on the mid-insula pattern) were highly correlated with temperature levels ($r_{196}$ = 0.454, p<0.001) as well as individual pain ratings ($r_{196}$ = 0.440, p<0.001). Together with the predictions using NS and FE vicarious pain-predictive patterns separately (*Figure 7—figure supplement 1*), our results demonstrate that the vicarious pain patterns respond to self-experienced somatic pain, confirming that the vicarious pain patterns reflect pain-associated information. All findings remained significant after correcting multiple comparisons via Bonferroni correction.

## A Meta-analytic decoding of the general vicarious pain-predictive pattern

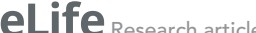

## B The general vicarious pain-predictive pattern (FDR *q* < 0.05)

**Figure 6.** A general vicarious pain-predictive pattern which predicts both observation of noxious stimulation of body limbs and facial expressions of pain induced vicarious pain. (A) Word cloud showing the top 50 relevant terms (excluding anatomical terms) for the meta-analytic decoding of the general vicarious pain-predictive pattern. The size of the font was scaled by correlation strength. (B) When thresholded at FDR *q* < 0.05, two-tailed

*Figure 6 continued on next page*

*Figure 6 continued*

(bootstrapped 10,000 samples) the general vicarious pain-predictive pattern revealed a distributed network of vicarious pain empathy representation including bilateral mid-insula and ventromedial prefrontal cortex.

## Discussion

Several studies have explored the neural underpinnings of vicarious pain in humans and suggested overlapping univariate fMRI activations in the anterior cingulate and insular cortices across different vicarious pain induction procedures (for meta-analyses see e.g. *Jauniaux et al., 2019*; *Timmers et al., 2018*). However, the conventional univariate approach lacks anatomical and functional specificity to test the question of whether vicarious pain across different modalities share common and process-specific neural representations (*Iannetti et al., 2013*; *Krishnan et al., 2016*; *Woo et al., 2014*; *Zaki et al., 2016*). Here, we employed a fine-grained MVPA approach which is sensitive and specific to particular types of mental processes including pain (*Kragel et al., 2018*; *Peelen et al., 2006*; *Wager et al., 2013*; *Woo et al., 2017*) to explore (1) whether shared neural representations of vicarious pain can be determined across different induction procedures (FE, NS) and (2) whether the shared neural representation is sensitive to pain-unspecific components of the vicarious pain response (arousal, negative affect) and related to the experience of somatic pain. We demonstrated that shared multivariate patterns encoding NS and FE vicarious pain can be determined at the whole-brain level and that across different analytic approaches the mid-insular cortex was consistently engaged across induction procedures. Furthermore, we demonstrated that these patterns were not sensitive to respond to the processing of non-painful high-arousal negative stimuli in the same sample, together with the findings showing that NS vicarious pain predicted FE vicarious pain (and vice versa) more accurately as compared with the predictions using a negative emotion

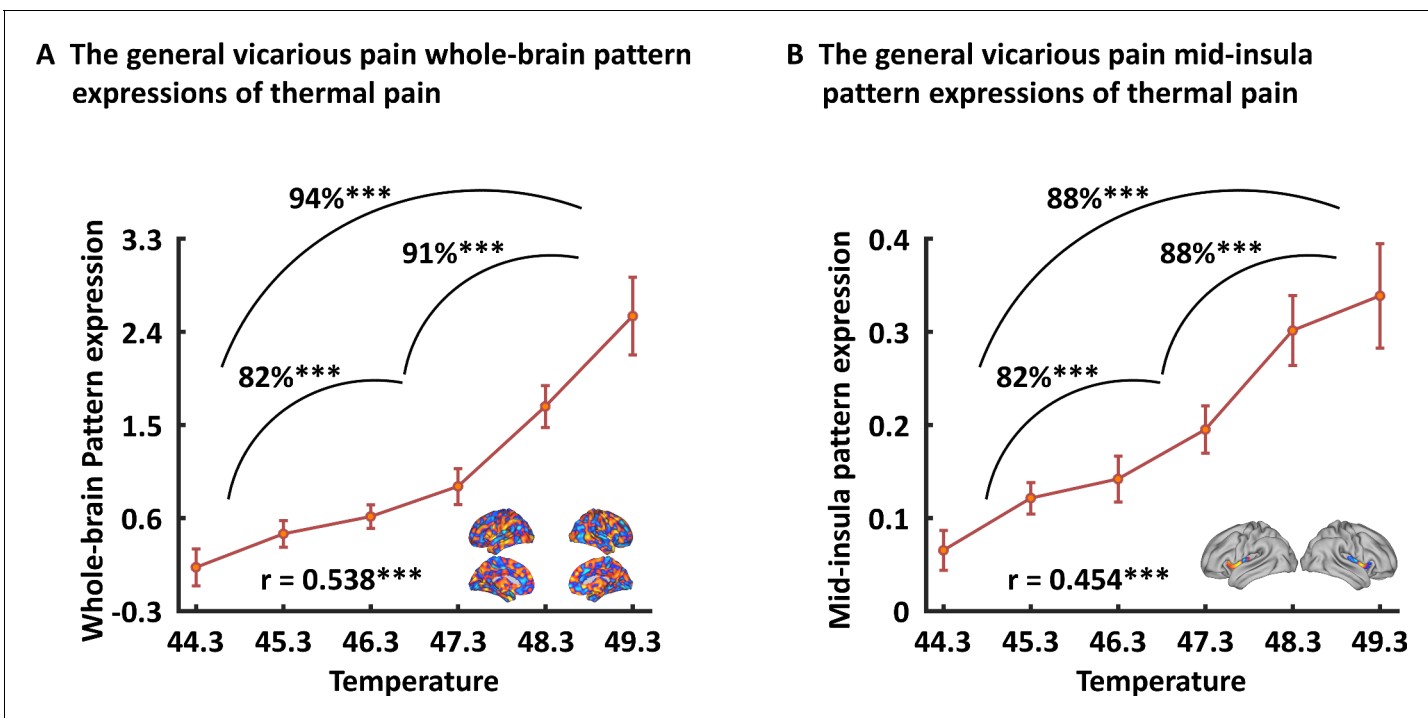

**Figure 7.** Generalizability of the general (across NS and FE) vicarious pain-predictive pattern. Both whole-brain (**A**) and mid-insula (**B**) vicarious pain-predictive patterns could accurately predict the severity and classify the levels of self-experienced pain in an independent dataset. ***p < 0.001. Error bar indicates standard error.

The online version of this article includes the following figure supplement(s) for figure 7:

**Figure supplement 1.** Generalizability of the NS and FE vicarious pain-predictive patterns.

**Figure supplement 2.** Varying sample size predictions.

decoder, suggesting that the common vicarious pain representations do not simply reflect shared unspecific processes of negative affect or arousal. Moreover, the shared vicarious pain representations predicted self-experienced thermal pain in an independent sample, suggesting an association between the neural expression and processes directly related to the experience of pain. Together these results provide evidence for a generalized neural representation of vicarious pain, particularly in the mid-insula, and demonstrated that the shared neural signature may specifically capture pain-associated aspects of the vicarious pain response rather unspecific processes such as aversive experience or arousal.

The idea that vicarious pain across different induction procedures share common neural representations has been supported by meta-analyses covering previous fMRI pain empathy studies that demonstrated overlapping activations in the insular and cingulate cortices (*Jauniaux et al., 2019*; *Timmers et al., 2018*). In line with these meta-analytic findings we found that these regions were consistently engaged during both NS and FE vicarious pain. However, the insular and anterior cingulate cortices are involved in a wide range of mental processes including representation of interoceptive and affective states as well as salience detection (*Craig, 2009*; *Critchley et al., 2004*; *Timmers et al., 2018*; *Uddin, 2015*), suggesting that the overlapping activity might be due to common underlying mental processes such as detecting and orienting attention toward salient stimuli or unspecific emotional arousal (*Corradi-Dell'Acqua et al., 2011*; *Corradi-Dell'Acqua et al., 2016*; *Valentini and Koch, 2012*).

To systematically test whether vicarious sharing of pain elicited by different social cues shares common neural representations, we developed and compared multivariate patterns that predicted NS and FE vicarious pain evoking stimuli respectively. While mass-univariate analysis results reflect the presence of intermingled neuronal populations related to stimulus-specific representations, MVPA investigates whether idiosyncratic spatial variations in the fMRI signal are shared or dissociated across different conditions and thus might be more suitable to determine process-specific representations in meso-scale neural circuits (*Kamitani and Tong, 2005*; *Kriegeskorte et al., 2006*; *Peelen and Downing, 2007*). Moreover, previous studies have suggested that whole-brain predictive models could better capture emotional processes compared to regional approaches, such as decoding of a single brain region or searchlight-based methods (*Kragel et al., 2018*; *Woo et al., 2017*). To this end, we first identified whole-brain fMRI patterns that accurately predicted NS and FE vicarious pain, respectively. We found that the NS and FE vicarious pain-predictive patterns were spatially correlated and both could classify within- and between-modality painful versus non-painful stimuli at the whole-brain level, suggesting that NS and FE vicarious pain share distributed processing across multiple systems and component processes. In line with previous studies demonstrating that while NS vicarious pain provides objective cues about the sensory component of the observed pain the FE vicarious lacks such information and is more subjective and indirect as the pain experience of the expresser need to be interpreted by the observer (*Hadjistavropoulos et al., 2011*; *Vachon-Presseau et al., 2012*), the decreased accuracies and effect sizes in the cross-modality predictions additionally suggest partly distinguishable neural representations of NS and FE vicarious pain possibly reflecting the engagement of different component processes.

In the context of previous studies suggesting that pain empathy deficits are mediated by regional-specific brain lesions and functional dysregulations (*Leigh et al., 2013*; *Shamay-Tsoory et al., 2009*; *Xu et al., 2020b*) the question for the contribution of specific brain regions arises. Thresholding the vicarious pain patterns (at FDR $q < 0.05$, two-tailed) allowed us to identify voxels that reliably contributed to the respective decoders and revealed that specifically the bilateral mid-insula provided important features to predict both NS and FE vicarious pain. Moreover, the mid-insula partial vicarious pain patterns were highly spatially correlated and both could significantly predict within- and between-modality vicarious pain-related experience. Consistent with this, searchlight-based classification analyses also demonstrated that mid-insula local patterns produced significant within- and between-modality predictions of vicarious pain. Our results are in line with a previous meta-analysis showing that the mid-insula responds specifically to empathy for pain across different task paradigms compared to empathy for non-pain negative affective states (*Timmers et al., 2018*), which together with the present findings suggests that the mid-insula represents a core neural substrate for vicarious pain.

Although multivariate predictive models can capture information at much finer spatial scales and consequently anatomical specificity (*Kamitani and Tong, 2005*; *Woo et al., 2017*), the question of

the specific mental processes captured by our vicarious pain-predictive patterns remains unclear. Pain empathy is a multi-component process that includes the vicarious sharing of pain but may also evoke emotional arousal and negative affect, and these unspecific processes can be captured by the decoders. To determine the functional specificity of the neural representations, we applied the vicarious pain-predictive patterns to data from an emotion processing paradigm acquired in the same sample as well as to data from a thermal pain induction experiment in an independent sample and found that (1) the vicarious pain patterns performed only modest for discriminating high-arousal (non-painful) negative stimuli from low-arousal neutral stimuli and (2) the whole-brain and mid-insula patterns predicted levels of self-experienced thermal pain with high accuracies. Finally, we developed a general vicarious pain-predictive pattern across NS and FE vicarious pain induction procedures and demonstrated that it accurately predicted both NS and FE vicarious pain (accuracies > 87%) as well as thermal pain intensities (accuracies > 82%), yet classified non-painful negative versus neutral stimuli with comparably low accuracy (59%). In line with the prediction results, meta-analytic decoding analysis revealed that this general vicarious pain pattern was highly correlated with the domains of 'painful' and 'pain', but not with 'arousal', 'valence' or 'negative' (not shown in the top 100 relevant terms). Together these findings suggest a shared neural representation of vicarious pain and a high-specificity of the whole-brain and specifically the mid-insula patterns for the vicarious experience of pain. A previous study developed a vicarious pain signature (VPS) that was sensitive and specific to NS vicarious pain, but not sensitive to the intensity of self-experienced somatic pain (*Krishnan et al., 2016*). Examining similarities with our general vicarious pain-predictive pattern revealed only modest spatial correlations between the two patterns (r = 0.04). The different instructions employed in the experiments might have contributed to the low overlapping spatial distributions such that participants in the previous study were required to explicitly rate their emotional response to the stimuli, whereas we decided for an implicit processing (passive viewing) paradigm to match instructions across the vicarious pain and negative emotional processing paradigm and to additionally control for cognitive processes which can modulate empathic reactivity and painful experience as well as the specific neural networks engaged (*Jauniaux et al., 2019*; *Urien and Wang, 2019*). Moreover, we found that the present pattern could successfully predict pain experience during thermal heat stimulation while the VPS was not sensitive to self-experienced pain. The observed differences might be explained in terms of (1) the considerably larger sample size included in the present study and prediction accuracy (as reflected by prediction-outcome correlation) of self-experienced pain experience increased as a function of sample size used to develop the NS vicarious pain decoder (see additional analysis presented in *Figure 7—figure supplement 2*) and (2) differences between paradigms and instructions such that, for example, a recent meta-analysis of pain empathy studies showed that the mid-cingulate gyrus was more activated by explicit cognitive/evaluative paradigms while the right inferior frontal gyrus and anterior insula were more activated by implicit perceptual/affective paradigms (*Timmers et al., 2018*).

Our results highlighted the mid-insula as a key region sharing similar neural representations across NS and FE vicarious pain suggesting that it may contribute to the core vicarious pain experience that characterizes pain empathy. Consistent with the whole-brain results, the shared information in the mid-insula was specific to vicarious pain rather than negative affect or arousal. Previous non-human primate and human studies indicate that the posterior and mid-insula receive nociceptive information from thalamic nuclei (*Craig et al., 1994*; *Craig et al., 2000*) which are in turn conveyed to the anterior insula for progressive integration with higher level affective and interoceptive experience (*Corradi-Dell'Acqua et al., 2011*; *Corradi-Dell'Acqua et al., 2016*; *Singer et al., 2009*). Although overarching models of the neural basis of pain empathy and neuroimaging meta-analyses (*Jauniaux et al., 2019*; *Timmers et al., 2018*) emphasize the role of the anterior insula in pain empathy processing, accumulating evidence from studies examining shared and process-specific representations of vicarious pain suggest a specific role of the mid-insula in vicarious pain (*Corradi-Dell'Acqua et al., 2011*; *Krishnan et al., 2016*), whereas the (left) anterior insula also responded to negative stimuli in general (*Corradi-Dell'Acqua et al., 2011*) and across modalities (*Corradi-Dell'Acqua et al., 2016*). Importantly, the peak anterior insula coordinates identified in these previous studies did not overlap with our mid-insula mask or the mid-insula region that exhibited reliable predictive features in both NS and FE vicarious pain whole-brain patterns determined in the present study, suggesting a more specific role of the mid-insula in pain-related components of the vicarious pain response (see also recent meta-analysis by *Timmers et al., 2018* demonstrating a specific role

of the mid-insula in pain empthy). In support of our findings a previous study employed a similar whole-brain MVPA approach to predict NS vicarious pain induced by an evaluative paradigm also identified the bilateral mid-insula as reliable (q < 0.05, FDR corrected) predictive regions (*Krishnan et al., 2016*), further conforming the reliable contribution of this region in encoding vicarious pain. Studies examining the functional and anatomical organization of the insular cortex with intracerebral electrical stimulation have demonstrated that painful sensations can be elicited by stimulation of the middle but not the anterior insula (*Afif et al., 2010*). Together with the functional relevance of the mid-insula to predict objective and subjective pain experience in an independent sample and the contribution of this region to nociception as well as vicarious pain (*Botvinick et al., 2005*; *Krishnan et al., 2016*; *Lamm et al., 2011*; *Timmers et al., 2018*; *Wager et al., 2013*), our findings suggest that the shared representations in the mid-insula across vicarious pain induction procedures may specifically code the automatic pain sharing which resonates with embodies conceptualizations of vicarious pain (see e.g. *Corradi-Dell'Acqua et al., 2011* for a convergent interpretation). However, consistent with previous evidence that (NS) vicarious pain representation is distributed across brain regions and single local regions exhibit considerably lower effect sizes compared to whole-brain predictive models (*Krishnan et al., 2016*), we found that the prediction effect sizes for the mid-insula were smaller than those observed in our whole brain analyses. These findings suggest that despite the key role of the mid-insula in vicarious pain experience this region is not sufficient to fully capture this process.

Consistent with previous studies suggesting that the anterior cingulate cortex represents a core brain region for emotional empathy in general and pain empathy in particular (*Fan et al., 2011*; *Jauniaux et al., 2019*; *Timmers et al., 2018*), we found overlapping deactivations in the rostral and ventral anterior cingulate cortex in mass-univariate analyses and shared patterns in the dorsal, rostral and ventral anterior cingulate cortex in searchlight-based prediction analyses between NS and FE vicarious pain. However, no overlapping reliable predictive voxels for whole-brain NS and FE pain-predictive patterns were found in cingulate regions suggesting a differential involvement of this region during FE and NS vicarious pain induction procedures. From a methodological perspective, these results may reflect that the whole-brain predictive model could provide a more specific neural description of a behavior or mental process (*Kragel et al., 2018*; *Woo et al., 2017*). In line with our findings, previous studies also showed that significant activation and searchlight-based prediction in local regions do not necessarily imply reliable predictive features in whole-brain predictive models (*Krishnan et al., 2016*; *Woo et al., 2014*). From a brain systems perspective, these findings may indicate that the anterior cingulate cortex is not specifically involved in vicarious pain elicited across induction procedures. Although the anterior cingulate cortex has been reliably identified in meta-analytic studies covering brain activation patterns during (pain) empathy induction procedures (see e.g. *Jauniaux et al., 2019*; *Timmers et al., 2018*), the anterior cingulate has also been associated with a number of basal processes, including arousal and salience, and activation in this region may reflect rather unspecific neural responses.

The present study has limitations that should be addressed in future studies. Compared to the homogeneous stimuli within the conditions of the vicarious pain and the self-experienced pain paradigm the stimuli displaying emotional evocative scenes from the IAPS database may have led to a higher inter-trial variance in the negative processing experiment. Although the inter-stimulus variance should not systematically differ between the experimental conditions employed to develop the corresponding decoder, we cannot fully exclude that this may have partly contributed to the low accuracies of the emotional processing decoder with respect to predicting self-experienced pain ratings. Moreover, the current study employed a passive observation paradigm and a recent meta-analysis revealed that vicarious pain induced by cognitive/evaluative and affective/perceptual paradigms elicited activations in overlapping yet also different brain regions (*Timmers et al., 2018*). Whether the present conclusions could generalize to more 'active' engagements in empathy (e.g. explicitly asking subjects to imagine that the injury occurring in the picture displayed was happening to them) remains to be determined.

In conclusion, by applying a novel whole-brain as well as local-region-based MVPA approaches in a large sample of healthy adults, our results provide the first neuroimaging evidence that NS and FE vicarious pain share common neural representations, especially in the mid-insula which may specifically encode the vicarious sharing of pain that specifically characterizes pain empathy. Moreover, we also provide a general vicarious pain-predictive pattern (across NS and FE vicarious pain stimuli),

which may be employed in future studies to facilitate inferences about pain empathy across modalities as well as self-experienced pain. Our study offers a new approach to better understand pain empathy by exploring common neural representations and linking these shared representations to felt pain.

# Materials and methods

## Key resources table

| Reagent type (species) or resource | Designation | Source or reference | Identifiers | Additional information |
|---|---|---|---|---|
| Software, algorithm | Matlab R2015b | MathWorks | RRID:SCR 001622 | |
| Software, algorithm | SPM12 | Wellcome Trust Centre for Neuroimaging | RRID:SCR_007037 | |
| Software, algorithm | CANLab Core Tools | CANlab | https://github.com/canlab | |
| Other | Thermal pain data | *Wager et al., 2013* | https://ndownloader.figshare.com/files/12708989 | |
| Other | Vicarious pain signatures | This paper | https://neurovault.org/collections/6332/ | Deposited multivariate patterns |
| Other | Data and codes | This paper | https://figshare.com/articles/Vicarious_pain_dataset/11994498 | Deposited fMRI data and scripts for figures |

## Participants

N = 252 healthy young participants were enrolled in the current study and underwent a previously validated NS and FE vicarious pain empathy fMRI paradigm. The fMRI data on the basic group activation maps for NS and FE vicarious pain contrasts were previously published in a study examining dimensional associations with trait autism and alexithymia (*Li et al., 2018*) and a study investigating network-level communication during pain empathic processing using an exploratory inter-subject phase synchronization approach (*Xu et al., 2020a*). Of note, the aim, methodological approach and hypotheses of the current study were independent from these previous publications; here, we focus on identifying an fMRI multivariate pattern for NS and FE vicarious pain separately and assessing their relationship. To further examine the specificity of the determined pain patterns from general negative emotion processing the data from an emotion processing paradigm from the same subjects was additionally used. Due to technical issues during data acquisition (incomplete data, n = 6), left-handedness (n = 4) or excessive head motion (>3 mm translation or 3° rotation; n = 4) data from 14 participants were excluded leading to a sample of n = 238 participants (118 females; mean ± SD age=21.58 ± 2.32 years) for the pain empathy analyses; data from 15 participants (incomplete data n = 8; left-handedness, n = 4; excessive head motion, n = 3) was excluded from the emotion processing paradigm analyses (n = 237; 120 females; mean ± SD age=21.55 ± 2.30 years). Participants provided written consent, the study was approved by the ethics committee at the University of Electronic Science and Technology and was in accordance the Declaration of Helsinki. Consent authorization for publication has been obtained from individuals in *Figure 1*.

## Experimental stimuli

The main aim of the present study was to determine (1) shared neural representations of pain empathy and (2) to further differentiate the specificity of the neural representation of shared vicarious pain from unspecific arousal and negative processing. For aim (1) we employed two different sets of validated pain empathy experimental stimuli displaying noxious stimulation of body limbs (NS vicarious pain) and facial expressions of pain (FE vicarious pain) as well as respective non-painful control stimuli (see *Figure 1A* for examples). The NS vicarious stimuli displayed a person's hand or foot in painful or non-painful everyday situations from the first-person perspective (e.g. the painful stimulus displays cutting a hand with a knife whereas the matched non-painful control stimulus shows cutting vegetables with a knife; for an evaluation of the stimuli see also *Meng et al., 2012*). The FE vicarious

stimuli incorporated painful and neutral facial expressions from 16 Chinese actors (eight males; for an evaluation of the stimuli see also *Sheng and Han, 2012*). To further validate the stimulus properties, we recruited an independent sample of 40 subjects (two of them were excluded due to incomplete data; 17 females; Mean ± SD age=20.45 ± 1.43 years) to rate the intensity of pain the depicted person is experiencing, the intensity of (vicarious) pain they experience while seeing the picture, valence and arousal for each stimulus on nine-point Likert scales (1 = 'not painful at all', 'very negative' or 'lowly arousing', 9 = 'extremely painful', 'very positive' or 'highly arousing'). In line with previous studies employing these stimulus sets (*Meng et al., 2012*; *Sheng and Han, 2012*) ratings in the present sample confirmed that both sets of painful stimuli were rated as considerably more painful in terms of the perceived level of pain the person in the picture is experiencing as well as level of vicarious pain experience in the observer (all *P*s <0.001). As expected, both sets of painful stimuli were also rated as more negative and stronger arousing than the control stimuli (all *P*s <0.001) (details see *Figure 1B* and Results). To determine whether the shared higher arousal and negative affect of both painful stimuli relative to their control stimuli may have contributed to the identified shared neural representation (aim 2) we additionally employed a stimulus set with non-painful high-arousal negative pictures and low-arousal neutral control stimuli. All stimuli were from the International Affective Picture System (IAPS) database. We recruited another independent sample of 37 subjects (16 females; Mean ± SD age=23.60 ± 2.86 years) to rate the valence and arousal for each stimulus with nine-point Likert scales (1 = 'very negative' or 'lowly arousing', 9 = 'very positive' or 'highly arousing'). Given that the IAPS stimuli we selected were non-painful we did not ask subject to rate pain intensity. Negative stimuli elicited substantial negative affect and arousal on numerical rating scales as compared with neutral stimuli (details see Results).

## Presentation of the stimuli

The pain empathy paradigm employed a blocked design incorporating condition-specific blocks presenting the validated visual stimuli displaying painful everyday scenes (NS vicarious pain) and painful facial expressions (FE vicarious pain) as well as modality-specific control stimuli displaying non-painful scenes (NS control) or neutral facial expressions (FE control). A total of 16 blocks (four blocks per condition) were presented in a pseudo-randomized order and interspersed by a jittered red fixation cross (8, 10, or 12 s). Each block (16 s) incorporated four condition-specific stimuli (each presented for 3 s) separated by a white fixation cross (1 s). An implicit processing paradigm (passive viewing) was employed. To this end, participants were instructed to attentively watch the presented stimuli.

In line with the pain empathy paradigm, the emotion processing paradigm employed a block design incorporating three experimental conditions (positive, negative and neutral pictures). A total of 19 blocks (neutral, seven blocks; negative, six blocks; positive, six blocks) were presented in a pseudo-randomized order and interspersed by a jittered red fixation cross (8, 10, or 12 s). In each block (16 s), four condition-specific stimuli (3 s) were presented and separated by a white fixation cross (1 s). An implicit processing (passive viewing) paradigm was employed and participants were asked to attentively watch the stimuli. To ensure attentive processing, participants were required to press a button when a stimulus with a white frame (one in each block) was presented.

## Thermal pain paradigm

Thirty-three healthy (22 females; mean ± SD age=27.9 ± 9.0 years), right-handed subjects participated in the thermal pain study (details see *Wager et al., 2013*; *Woo et al., 2015*). Six levels of temperature (ranging from 44.3°C to 49.3°C in increments of 1°C) were delivered to the volar surface of the left inner forearm using a TSA-II Neurosensory Analyzer (Medoc Ltd.) with a 16 mm Peltier thermode end-plate during fMRI acquisition. The fMRI task included seven passive experience runs and two regulation runs where subjects were asked to cognitively 'increase' (regulate-up) or 'decrease' (regulate-down) pain intensity with each run encompassing 11 trials. Each trial consisted of a 12.5 s stimulus (3 s ramp-up and 2 s ramp-down periods and 7.5 s at the target temperature), a jittered 4.5–8.5 s delay, a 4 s painful/non-painful decision period, a 7 s continuous warmth or pain rating period (on a visual analogue scale) and 23–27 s rest. For the current study, we incorporated the data from the passive experience runs.

## MRI data acquisition and preprocessing

MRI data were collected on a 3.0 T GE Discovery MR750 system (General Electric Medical System, Milwaukee, WI). Functional MRI data was acquired using a T2*-weighted echo-planar imaging (EPI) pulse sequence (repetition time = 2 s, echo time = 30 ms, 39 slices, slice thickness = 3.4 mm, gap = 0.6 mm, field of view = 240 × 240 mm, resolution = 64 × 64, flip angle = 90°, voxel size = 3.75 × 3.75 × 4 mm). To improve spatial normalization and exclude participants with apparent brain pathologies a high-resolution, T1-weighted image was acquired using a 3D spoiled gradient recalled (SPGR) sequence (repetition time = 6 ms, echo time = minimum, 156 slices, slice thickness = 1 mm, no gap, field of view = 256 × 256 mm, acquisition matrix = 256 × 256, flip angle = 9°, voxel size = 1 × 1×1 mm). OptoActive MRI headphones (http://www.optoacoustics.com/) were used to reduce acoustic noise exposure for the participants during MRI data acquisition.

Functional MRI data was preprocessed using Statistical Parametric Mapping (SPM12; RRID:SCR_007037; https://www.fil.ion.ucl.ac.uk/spm/software/spm12/). The first 10 volumes of each run were discarded to allow MRI T1 equilibration and active noise cancelling by the headphones. The remaining volumes were spatially realigned to the first volume and unwarped to correct for nonlinear distortions related to head motion or magnetic field inhomogeneity. The anatomical image was segmented into grey matter, white matter, cerebrospinal fluid, bone, fat and air by registering tissue types to tissue probability maps. Next, the skull-stripped and bias corrected structural image was generated and the functional images were co-registered to this image. The functional images were subsequently normalized the Montreal Neurological Institute (MNI) space (interpolated to 2 × 2 × 2 mm voxel size) by applying the forward deformation parameters that were obtained from the segmentation procedure, and spatially smoothed using an 8 mm full-width at half maximum (FWHM) Gaussian kernel.

## Pain empathy - univariate general linear model (GLM) analyses

A two-level random effects GLM analysis was conducted on the fMRI signal to determine shared modality-specific activation patterns using a mass-univariate GLM approach. The first-level model included four condition-specific (NS vicarious pain, NS control, FE vicarious pain, and FE control) box-car regressors logged to the first stimulus presentation per block that were convolved with SPM12's canonical hemodynamic response function (HRF). The fixation cross epoch during the inter-block interval served as implicit baseline, and a high-pass filter of 128 s was applied to remove low-frequency drifts. Regressors of non-interest (nuisance variables) included (1) six head movement parameters and their squares, their derivatives and squared derivatives (leading to 24 motion-related nuisance regressors in total) and (2) motion and signal-intensity outliers (based on Nipype's rapidart function). Single-subject voxel-wise statistical parametric maps for the empathy modality-specific contrasts (NS vicarious pain >NS control and FE vicarious pain >FE control) were obtained and subjected to group-level one-sample t-tests. The corresponding analyses were thresholded and corrected for multiple comparisons within a grey matter mask based on false discovery rate (FDR $q <$ 0.05, two-tailed) with a minimum extent of 100 mm$^3$. The resulting thresholded activation maps were next used to identify common regions of activation across the modalities (NS and FE vicarious pain; i.e. masking the overlapping significant voxels).

To determine the activation similarity of NS and FE vicarious pain, a permutation-based correlation analysis was employed (*Hong et al., 2019*). Specifically, we (1) calculated Pearson's correlation (r) between the modality-specific unthresholded statistical maps (NS vicarious pain >NS control versus FE vicarious pain >FE control), (2) shuffled the condition labels for the NS stimuli, obtained a new group-level statistical map for 'NS vicarious pain >NS control' and calculated the activation similarity of FE and the 'modelled' NS vicarious pain, (3) repeated step (2) 10,000 times, (4) repeated steps (2-3) with shuffled labels for FE instead of NS stimuli, and finally (5) calculate the probability of observing the activation similarity between the true NS and FE pain given the null distribution of permuted activation similarity. A p value < 0.05 was being considered statistically significant and between 0.05 and 0.1 was being considered as marginal significant.

## Pain empathy - multivariate pattern analyses

For the multivariate pattern analyses, nuisance regression (24 head motion parameters, motion and signal-intensity outliers, and linear trend) and high-pass filtering (cut off at 128 s) were initially

simultaneous conducted on the preprocessed fMRI data. Next, the fMRI signal was averaged within the four condition-specific blocks (shifted by 3 TRs to account for the delay of the HRF). In line with previous studies (e.g. *Krishnan et al., 2016*; *Wager et al., 2013*; *Woo et al., 2014*), we used normalized and smoothed (8 mm FWHM Gaussian kernel) data to develop the population-level vicarious pain-predictive patterns as previous studies suggested that this smoothing level could improve intersubject functional alignment while retaining sensitivity to mesoscopic activity patterns that are consistent across subjects (*Op de Beeck, 2010*; *Shmuel et al., 2010*). Linear support vector machines (SVMs, C = 1) were then employed to the whole-brain maps (restrict to a grey matter mask) to train multivariate pattern classifiers on the cleaned averaged fMRI signal to discriminate NS vicarious pain versus NS control and FE vicarious pain versus FE control separately. The classification performance was evaluated by a 10-fold cross-validation procedure during which all participants were randomly assigned to 10 subsamples of 23 or 24 participants using MATLAB's cvpartition function. The optimal hyperplane was computed based on the multivariate pattern of 214 or 215 participants (training set) and evaluated by the excluded 24 or 23 participants (test set). The training set was linearly scaled to [−1, 1], and the test set was next scaled using the same scaling parameters before applying SVM (*Hsu et al., 2003*). This procedure was repeated 10 times with each subsample being the testing set once. To avoid a potential bias of training-test splits, the cross-validation procedures throughout the study were repeated 10 times by producing different splits in each repetition and the resultant accuracy and p values were averaged to produce a convergent estimation (*Zhou et al., 2018*). In line with the mass-univariate analyses and to identify which brain regions made reliable contributions to the decoders (*Wager et al., 2013*; *Zhou et al., 2019*), the pattern maps were thresholded at FDR $q < 0.05$ (two-tailed) with a minimum extent of 100 mm$^3$ using bootstrap procedures with 10,000 samples. Next the thresholded maps were subjected to a conjunction analysis to identify regions that robustly contributed to both NS and FE vicarious pain classifiers by masking overlapping significant voxels. Statistical maps were visualized using the Connectome Workbench provided by the Human Connectome Project (https://www.humanconnectome.org/software/connectome-workbench).

Similarity patterns between the modality-specific neural patterns were determined employing (1) Pearson's correlation between the whole-brain unthresholded classifier weights using a permutation test (similar to the activation similarity analysis) and (2) 'between - modality classification' tests encompassing the following two steps: (a) pattern classifiers were trained separately for NS vicarious pain versus NS control and FE vicarious pain versus FE control with a 10-fold cross-validation procedure (repeated 10 times), and next (b) applying the identified patterns of NS and FE vicarious pain to out-of-sample participants for the FE vicarious pain versus FE control and NS vicarious pain versus NS control respectively using a two-alternative forced choice test, where pattern expression values were compared for two conditions with the image exhibiting the higher expression being determined as pain.

## Pain empathy – within- and between- modality classification analyses employing local classifiers

To further identify regions with shared neural expressions across NS and FE vicarious pain, a local pattern-based classification approach with three-voxel radius spherical searchlights around center voxels was employed (*Corradi-Dell'Acqua et al., 2011*; *Kriegeskorte et al., 2006*; *Woo et al., 2014*). Specifically, (1) multivariate pattern classifiers using a defined local region were trained to discriminate vicarious pain versus control within each modality (i.e. NS and FE stimuli) separately and (2) the patterns obtained were next applied to out-of-sample participants for within-modality cross-validation and between-modality cross-prediction. Steps (1) and (2) were repeated for each local region across the whole-brain. It was hypothesized that shared neural representations for NS and FE pain within a local region would be reflected by significant cross-validation and cross-prediction accuracies for each classifier. Given that the specific results of searchlight-based approaches strongly depend on the searchlight size if information is not present and detected equally at all spatial frequencies (*Etzel et al., 2013*), we repeated our analyses with two additional searchlight sizes (4-mm- and 10-mm-radius spheres).

## Specificity of the NS and FE vicarious pain-predictive patterns

To test whether the observed NS and FE vicarious pain-predictive patterns were specific to pain processing or rather reflect general aspects of negative emotional processing, the two pain-predictive patterns were applied to the data from the emotional task paradigm. The first-level model for the emotion processing data included the four experimental conditions (positive, negative, neutral and white framed stimuli) and high-pass filter and nuisance regressors were identical to the pain empathy GLM analysis. The two pain-predictive patterns were next applied to negative and neutral contrasts (via dot-products) using a repeated 10-fold cross-validation procedure separately, and subsequently two-alternative forced choice tests were employed to discriminate negative versus neutral stimuli.

## Generalized vicarious pain-predictive pattern

Given that we found shared neural representations between NS and FE vicarious pain (see Results for details), a general vicarious pain pattern was developed by classifying vicarious pain (NS and FE) versus control stimuli and further evaluated by predicting NS vicarious pain versus NS control and FE vicarious pain versus FE control separately through 10-fold cross validation procedures. We next constructed 10,000 bootstrap sample sets to visualize the voxels that made the most reliable contribution to the classification and to decode the cognitive relevance of the classifier with the resultant Z map using the Neurosynth (*Yarkoni et al., 2011*). Moreover, to compare the general vicarious pain pattern with the NS and FE vicarious patterns, we examined the similarities between this general vicarious pain pattern and the NS and FE vicarious pain patterns, respectively.

## Generalizability of the vicarious pain pattern

To test the functional relevance and generalizability of the empathic-induced neural pain pattern, the unthresholded whole-brain pattern of the general across NS and FE vicarious pain was applied to determine the behavioral and neural responses during actual pain induction. To this end, data from a previous study employing different levels of thermal pain induction during fMRI scanning (MRI data acquisition and preprocessing details see *Wager et al., 2013*; *Woo et al., 2015*). First-level GLM analysis included regressors for stimulation periods for each of the six levels and the 11 s rating periods as well as nuisance regressors including intercept for each run, linear drift across time within each run, indicator vectors for outliers and head movement. The general vicarious pain pattern from the current study was used to estimate the pattern expressions of each participant in each condition (stimulation period) and next the neural pattern expressions of the six pain levels were (1) correlated with the temperature levels (1-6) as well as the subjective pain ratings separately and (2) employed to discriminate high thermal pain stimulation (average of 48.3°C and 49.3°C) versus low stimulation (average of 44.3°C and 45.3°C), high stimulation versus medium stimulation (average of 46.3°C and 47.3°C), as well as medium stimulation versus low stimulation. Moreover, we conducted the same analyses with NS and FE vicarious pain patterns to determine the robustness of the prediction.

## Data availability

Statistical and pattern weight images are available on Neurovault (https://neurovault.org/collections/6332/). Vicarious pain dataset as well as numerical data and Matlab scripts that were used to generate the figures are available on figshare (https://figshare.com/articles/Vicarious_pain_dataset/11994498). Other data can be obtained from the corresponding authors upon reasonable request.

## Code availability

Code is available at https://github.com/canlab (*Canlab, 2020*) and from the corresponding authors upon reasonable request.

## Acknowledgements

This work was supported by the National Key Research and Development Program of China (Grant No. 2018YFA0701400), National Natural Science Foundation of China (NSFC, No 91632117, 31700998, 31530032); Fundamental Research Funds for Central Universities (ZYGX2015Z002),

Science, Innovation and Technology Department of the Sichuan Province (2018JY0001); National Institute of Mental Health (R01 MH116026) and National Institute of Biomedical Imaging and Bioengineering (R01EB026549).

## Additional information

### Funding

| Funder | Grant reference number | Author |
| --- | --- | --- |
| National Natural Science Foundation of China | 91632117 | Benjamin Becker |
| National Natural Science Foundation of China | 31700998 | Keith M Kendrick |
| National Natural Science Foundation of China | 31530032 | Shuxia Yao |
| National Institute of Mental Health | R01 MH116026 | Tor D Wager |
| National Institute of Biomedical Imaging and Bioengineering | R01EB026549 | Tor D Wager |
| National Key Research and Development Program of China | 2018YFA0701400 | Benjamin Becker |
| Fundamental Research Funds for Central Universities | ZYGX2015Z002 | Benjamin Becker |
| Science, Innovation and Technology Department of the Sichuan Province | 2018JY0001 | Benjamin Becker |

The funders had no role in study design, data collection and interpretation, or the decision to submit the work for publication.

### Author contributions

Feng Zhou, Formal analysis, Visualization, Methodology, Writing - original draft, Writing - review and editing; Jialin Li, Data curation, Formal analysis, Writing - original draft; Weihua Zhao, Lei Xu, Xiaoxiao Zheng, Meina Fu, Data curation, Project administration; Shuxia Yao, Data curation, Project administration, Writing - review and editing; Keith M Kendrick, Conceptualization, Supervision, Project administration, Writing - review and editing; Tor D Wager, Resources, Software, Supervision, Methodology, Writing - review and editing; Benjamin Becker, Conceptualization, Funding acquisition, Writing - original draft, Project administration, Writing - review and editing

### Author ORCIDs

Feng Zhou (iD) https://orcid.org/0000-0002-4509-4026
Keith M Kendrick (iD) http://orcid.org/0000-0002-0371-5904
Benjamin Becker (iD) https://orcid.org/0000-0002-9014-9671

### Ethics

Human subjects: All participants provided written informed consent for study participation and consent to publish the data. The study and all procedures were approved by the local ethics committee at the University of Electronic Science and Technology of China (Approval ID: 298) and was in accordance with the most recent revision of the Declaration of Helsinki.

### Decision letter and Author response

Decision letter https://doi.org/10.7554/eLife.56929.sa1
Author response https://doi.org/10.7554/eLife.56929.sa2

## Additional files

**Supplementary files**

• Supplementary file 1. Table shows post-fMRI subjective ratings for vicarious pain evoking stimuli (Mean ± SD). NS vicarious pain, observation of noxious stimulation of body limbs induced vicarious pain; FE vicarious pain, observation of facial expressions of pain induced vicarious pain; NS control stimuli depict body limbs in similar but innocuous situations, FE control stimuli show neutral facial expressions.

• Transparent reporting form

### Data availability

The functional MRI, numerical data as well as the Matlab scripts used to generate the figures have been deposited on the figshare repository under accession code 11994498 (https://figshare.com/articles/Vicarious_pain_dataset/11994498) Statistical and pattern weight maps are available on the Neurovault repository under collection 6332 (https://neurovault.org/collections/6332/). Statistical and pattern weight images are available on Neurovault.

The following datasets were generated:

| Author(s) | Year | Dataset title | Dataset URL | Database and Identifier |
|---|---|---|---|---|
| Zhou F, Li J, Zhao W, Xu L, Zheng X, Fu M, Yao S, Kendrick KM, Wager TD, Becker B | 2020 | Vicarious pain dataset | https://figshare.com/articles/Vicarious_pain_dataset/11994498 | figshare, 10.6084/m9.figshare.11994498 |
| Zhou F, Li J, Zhao W, Xu L, Zheng X, Fu M, Yao S, Kendrick KM, Wager TD, Becker B | 2020 | Emotional contagion of pain across different social cues shares common and process-specific neural representations | https://neurovault.org/collections/6332/ | NeuroVault, 6332 |

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
