## [Decision Letter]

Thank you for submitting your article "Emotional contagion of pain across different social cues shares common and process-specific neural representations" for consideration by *eLife*. Your article has been reviewed by three peer reviewers, and the evaluation has been overseen by Alex Shackman (Reviewing Editor) and Christian Büchel (Senior Editor). The following reviewers have agreed to reveal their identity: Mitul Mehta (Reviewer #2); Deniz Vatansever (Reviewer #3).

Dr Shackman has drafted this decision to help you prepare a revised submission:

Summary

In this study, Zhou and colleagues have sought to determine shared neural representations in the processing of physical and affective vicarious pain in a large-scale fMRI study. In addition to the conventional univariate analysis, the authors employed extensive multivariate pattern analyses which highlighted common representations centred largely on the bilateral mid-insula. In a further validation step, the authors have shown that the identified patterns did not predict arousal in a separate emotional fMRI task within the same cohort, but predicted self-experienced pain in a thermal stimulation experiment within an independent sample. Utilising these results, the authors have also developed a domain-general pattern that could be employed in future studies.

The reviewers were enthusiastic about your report:

- Overall, this is an excellent study with a clear design and objective, accompanied by a well-written manuscript. I have no doubt that the results will be of interest to a wide audience in both the scientific and the general public.

- Multiple methods across different data sets, and also look at validation against task parameters and subjective responses.

- There are many strengths and positive things in this paper, above all the very large sample size, the use of complimentary analysis methods, and the attempt to make inferences stronger by bringing in data from several tasks/paradigms.

- The overall approach of univariate and multivariate methods with within and between modality classification is thoughtful and the authors show awareness of the potential broad role of key areas in the network and attempt to support specificity through the comparative analysis with the negative stimuli and a Neurosynth summary of the region.

- Strengths include the number of participants and the use of different analysis approaches and separate data sets and validation against task parameters.

- Overall, this is an excellent study with a clear design and objective, accompanied by a well-written manuscript. I have no doubt that the results will be of interest to a wide audience in both the scientific and the general public.

- Analyses were conducted and reported with the utmost scientific rigour.

Nevertheless, the 3 reviewers did have some suggestions for strengthening your report, as detailed below.

I would like to thank them for their constructive suggestions and their investment in the paper and the process.

Major/Moderate Suggestions

1) Need for Clear and Precise Nomenclature; Jingle-Jangle Issues.

a) A reviewer noted, "Throughout the manuscript, the authors use "emotional contagion" to refer to vicarious pain, which is slightly confusing given that the results indicate how pain representations are separate from emotional arousal. Hence, empathic contagion or some other term which does not refer to "emotion" might be more suitable.

b) Another reviewer wrote that, "We do not have enough clarity about what conceptual difference or connection exists between the two paradigms, and thus how we should interpret the neural differences; someone who focuses on facial emotion perception might thus rather conclude that emotion expressions are preferentially processed in the mid insular cortex "

c) S/he went on to note that, "I do not follow why the authors speak of a physical and an affective vicarious pain, and that the latter should measure/tap into emotion contagion; what I see are two paradigms showing (possibly) the same type of event happening to a target person (pain), one being triggered by a (mild) physical injury, the other by expressing pain on the face, likely expressed in response to a similar event, or not (this doesn't become clear, but neither it was probably to participants, I am assuming that because the paradigms were labeled as passive observation); in order to label one as "emotion contagion" and the other as something else (it remains unclear what that would be), one would need to clearly define emotion contagion first, and then have some independent measures of that it occurs; e.g. increase in heart rate, matching facial expressions, etc. moreover, if the task was presented in a passive-viewing fashion without any specific instruction (see 2), subjects may have associated the different kinds of stimuli with each other in such an experimental context: Intense somatic pain will always (in healthy individuals) elicit a strong affective reaction, and on the other hand, intense facial expressions of pain are most likely associated with physical pain. To play devil's advocate, one could also label the two paradigms as "passive pain perception via facial expression" vs. "passive pain perception via somatosensory events"

2) Insufficient motivation for focusing on mid-insula

a) A reviewer notes that "The authors base their mid-insula-restricted analysis ("Shared representations in the mid-insula") of shared representations on a) their own findings within the same manuscript and b) one paper reporting a specific role of the middle portion of the insula. Judging from the presented maps in Figure 3D, activity in the insula stretches from anterior to posterior portions. Also, e.g. Corradi-dell'Acqua et al., 2016 report a specific role of the anterior insula in vicarious pain. Authors need to justify this approach more clearly (or expand their ROI), especially because it could clearly affect the outcome of other analyses, such as the mid-insula classification approach for the IAPS paradigm."

3) Additional Analyses? One of the reviewers wondered whether the authors have measured affect in the pain fMRI task besides emotional arousal, and whether the identified pattern would relate to how positive/negative the participants found the given pain stimuli. This might provide evidence that the results could be separated from general emotional arousal.

4) Additional Analyses? The authors show that the patterns in pain task could not predict ratings in the emotional task. Is it possible to test whether that the reverse pattern is evident? (i.e. the pattern in the emotional task does not predict responses in the pain task.)

5) Framing/Discussion of Differences

a) Authors need to crisply articulate how the significance of observed differences in light of the many perceptual and psychological differences across tasks.

b) For example, a reviewer notes that, "The matching between the two vicarious conditions, but also to the control or comparison conditions, is far from optimal, so that many differences as well as a lack of "sharedness"/common variance between paradigms could be simply explained by that. In one case, the "neutral" condition is not fully neutral, as the picture shows a potentially noxious object that could also be seen as "just having missed to inflict pain/injury"; in the other, a painful expression is compared to a fully neutral one, without a noxious object; thus (visual and affective) salience between conditions starkly differs, which makes it hard to compare them and to draw meaningful conclusions when doing so;"

c) S/he notes that, "things get even more problematic when comparing data from the IAPS and the thermal pain conditions, which differ in many more aspects. In the IAPS, pictures of various emotionally evocative scenes are shown, thus some of the differences may be explained by higher intertrial variance and thus also lower "SNR" in the IAPS runs; moreover, salience might have differed as well – so any difference between conditions might be related to that, and the conclusion "it is not just emotional arousal that triggers the vicarious pain responses" is not conclusive (although I would be very much in favor of it); what could have been done to better match conditions would be e.g. to compare different facial emotion expressions matched for emotional arousal, so the "modality is not changed";

d) And, "the same limitation (about limited matching between conditions) applies to the comparison to the thermal pain condition, which is a very different condition in terms of trial structure, "immediateness" of experience for the participants, and visual stimulation (apart from the sample size)

6) Need for a sober and more complete discussion of implications, and perhaps an additional analysis

a) A reviewer notes that "the vicarious pain pattern successfully distinguishes between different thermal pain levels, speaking for shared neural patterns between self-experienced pain and vicarious pain, which appears to be the most interesting (yet deemphasized) finding of the study. Yet, after repeatedly emphasizing the advantages of MVPA over mass-univariate techniques, the authors argue that "although the vicarious pain patterns could accurately predict the intensity of self-experienced pain this result does not imply that vicarious and somatic pain share common neural representations", without further elaboration. Dismissing this finding in such a way leaves the manuscript with the sole main finding that physical and affective vicarious pain show overlap also on a multivariate pattern level; thus what seems crucial is to carve out much better the discrepancy between the current results and the previous ones by Krishnan in *eLife* as well; as a reader, I am left wondering what is correct now – that there are, or there aren't "shared representations"? the interpretation "Krishnan et al. used different instructions" is of course one possible explanation, but what about sample size and power? Maybe Krishnan et al. was underpowered, and the shared patterns would have been found there as well? But then why should a passive observation paradigm be "better" in identifying shared patterns, if anything I would have expected the opposite as my own experience as well as prior work e.g. by Feng et al. (in NeuroImage) tell us that reduced attention if anything leads to less activation in insula, A/MCC, and the likes? (apart from the fact that a between sample prediction should also have less sensitivity). My proposal here would be to go much deeper into the discrepancy between the current and Krishnan's findings – that alone could be a very interesting paper, given the current sample size; one way to do that could be e.g. to "simulate" (e.g. by random drawing and permutation testing with N as in Krishnan) the lower sample size of Krishnan and to test whether or not the differences between studies remain."

b) S/he notes that, "the discrepancies between the uni and multivariate analyses are another "gold mine" of this paper, but they are hardly addressed or explained. For example, the effect size (r) of the univariare analyses in Figure 1 is very low (0.171 -> ~4% common variance), whereas the MVPA analyses suggest much stronger, up to ~50%, correspondence. The authors need to seriously grapple with this in the Discussion.

[Editors' note: further revisions were suggested prior to acceptance, as described below.]

Thank you for resubmitting your work entitled "Empathic pain evoked by sensory and emotional-communicative cues share common and process-specific neural representation" for further consideration by *eLife*. Your revised article has been evaluated by Drs. Büchel (Senior Editor) and Shackman (Reviewing Editor). Dr. Shackman writes…

I am very pleased to accept your paper for publication pending receipt of a revision that addresses the few remaining suggestions. I will not send it out for another round of external review.

The reviewers and I were enthusiastic about the revision:

– The authors have taken on board all of my comments and presented detailed responses and further analyses of the data. All of these provide assurances that the results are robust. The new figures and supplementary figures are great and illustrate the data and the robustness of the findings well.

– The authors have diligently responded to all my prior concerns and comments, and carried out the additional analyses I have suggested. I think the current version of this study/manuscript presents major advancement in the field both from theoretical and technical perspectives. I have no further comments.

– The authors have been very responsive to the reviewer comments, and I would like to thank them for that. The paper has much improved as a consequence… Again, congratulations on a fine and huge piece of work, that certainly will advance the field.

Nevertheless, they did have some suggestions for further enhancing the report.

"It would benefit the paper to a. delineate more clearly/accessibly, what the mid-insula entails (there is a reference to the GitHub and paper, which is already great, but not all readers will want to check), and how it "differs"/can be disambiguated from "anterior" and possibly also "posterior" IC (anterior has been often reported in empathy but also in salience research)"

I leave it to the authors to determine whether they wish to tackle the following additional analysis…

"It might be useful to also test whether these other divisions (AI, PI) show similar or distinct effects as the mid-insula."

Or, whether they would prefer to simply amend the manuscript to address this constructive suggestion:

"Basically, it would be good to know whether the mid-insula cluster includes previously reported anterior insula clusters (at least partially), and whether what we should take away really is a stronger focus on mid rather than anterior insula in future empathy/vicarious pain studies and models."

Again, I am comfortable with the authors eschewing additional analyses if they wish, but they need to address the substance of the reviewer's very reasonable suggestion prior to publication.

---

## [Author Response]

Major/Moderate Suggestions1) Need for Clear and Precise Nomenclature; Jingle-Jangle Issues.a) A reviewer noted, "Throughout the manuscript, the authors use "emotional contagion" to refer to vicarious pain, which is slightly confusing given that the results indicate how pain representations are separate from emotional arousal. Hence, empathic contagion or some other term which does not refer to "emotion" might be more suitable.

We fully agree with the reviewer that using the terms “emotional contagion” and “vicarious pain” interchangeably, and using them together with “emotional arousal” in this context, might not be specific enough to differentiate between the mental processes referred to. Moreover, recent overarching conceptualizations of empathy clearly separate contagion from empathic responses, at least in the respect that contagion lacks a clear self-other distinction (e.g., Bird and Viding, 2014; De Vignemont and Singer, 2006; Lamm et al., 2019). We thank the reviewer for raising this important issue and included a clearer definition of the shared mental process in the Introduction as follows and changed the corresponding wording throughout the revised version accordingly to “vicarious pain”:

“…Vicarious pain can be triggered by observing or imagining another individual’s painful state and can be elicited by multiple types of social cues, particularly the observation of an inflicted physical injury or a facial expression of pain (Decety and Ickes, 2009; Jauniaux et al., 2019; Vachon-Presseau et al., 2012; Yesudas and Lee, 2015). While stimulus depicting the noxious stimulation of body limbs [i.e., observation of noxious stimulation induced vicarious pain (NS vicarious pain)] provides objective cues about the sensory component of the observed pain, the observation of facial expressions of pain [i.e., facial expressions induced vicarious pain (FE vicarious pain)] is considered more subjective and indirect as the pain experience of the expresser need to be interpreted by the observer (Hadjistavropoulos et al., 2011; Vachon-Presseau et al., 2012)… Despite the different psychological domains engaged in the pain empathic response induced by NS and FE both elicit vicarious pain experience (Timmers et al., 2018), encompassing pain-specific processes such as recognizing and understanding the painful state of the other person and affective sharing of pain but also non-specific processes that are shared between pain and other non-painful experiences such as arousal and negative affect (Zaki et al., 2016).”

b) Another reviewer wrote that, "We do not have enough clarity about what conceptual difference or connection exists between the two paradigms, and thus how we should interpret the neural differences; someone who focuses on facial emotion perception might thus rather conclude that emotion expressions are preferentially processed in the mid insular cortex "

We thank the reviewer for raising this important point and agree that the conceptual background and rationale to include the different paradigms into the study needs to be clarified. For the difference and connection between the two sets of vicarious pain stimuli please see also our reply to comment #1a. The major aims of the present study were to determine (1) shared – rather than different – and specific neural representations of vicarious sharing of pain across two different sets of pain empathy evoking stimuli, (2) test the specificity of the shared neural representations (relative to general negative affect / arousal), and (3) determine associations with experienced somatic pain. The corresponding analytic approach thus tested (1) whether the vicarious pain signatures that were developed by two different sets of stimuli separately exhibit shared empathic and pain-specific neural representations (which might provide an initial evidence that vicarious pain across different pain empathy evoking stimuli share common representations or not), (2) whether a general vicarious pain signature which could generalize across different types of stimuli that elicit vicarious pain can be determined and (3) whether the direct experience of somatic pain could “activate” this general vicarious pain signature. To this end we trained and tested vicarious pain predictive patterns across (a) images of noxious stimulation of body limbs [observation of noxious stimulation induced vicarious pain (NS vicarious pain)], and (b) facial expressions of pain [observation of facial expressions of pain induced vicarious pain (FE vicarious pain)]. However, both sets of stimuli could also induce unspecific processes such as arousal and negative affect and the corresponding cross-modality representation could simply reflect these more general mental processes (Corradi-Dell’Acqua et al., 2016; Zaki et al., 2016). One approach to test the specificity of the neural representation is to elicit arousal and negative affect by other (non-painful) stimuli and determine whether the neural representation is sensitive to non-pain related induction of these mental processes (please see also our response to comment #5C).

Based on the comment from the reviewer we included a clearer description of the aims of the present study and the corresponding operationalization and analytic strategy in the Introduction. Moreover, to further determine the properties of the experimental stimuli we included new data from an independent sample who explicitly rated the pain empathic stimuli. Briefly, subjects were asked to rate (1) pain intensity experienced by the subjects displayed in the picture (“how much pain do you think the person in the photo is feeling”) assessed by a nine-point Likert scale ranging from “1 = not painful at all” to “9 = extremely painful”, (2) pain intensity experienced by the participant in response to the stimulus (“how much pain do you experience when watching the picture”) assessed using the same Likert scale ranging from “1 = not painful at all” to “9 = extremely painful”, (3) valence of the stimuli by means of a nine-point Likert scale ranging from “very negative” to “very positive”, and arousal of the stimuli by means of a nine-point Likert scale ranging from “very low arousing” to “very high arousing”. These ratings confirmed that relative to the respective control stimuli both sets of painful stimuli were perceived as more painful in terms of recognized and shared pain as well as more arousing and negative (details see Supplementary File 1 and Figure 1B). To control for the shared higher arousal and negative affect induced by both sets of pain-stimuli we thus incorporated a set of non-painful yet high-arousal negative stimuli from the IAPS database as well as corresponding low-arousal neutral pictures to induce arousal and negative affect independent of pain (empathy) aspects.

c) S/he went on to note that, "I do not follow why the authors speak of a physical and an affective vicarious pain, and that the latter should measure/tap into emotion contagion; what I see are two paradigms showing (possibly) the same type of event happening to a target person (pain), one being triggered by a (mild) physical injury, the other by expressing pain on the face, likely expressed in response to a similar event, or not (this doesn't become clear, but neither it was probably to participants, I am assuming that because the paradigms were labeled as passive observation); in order to label one as "emotion contagion" and the other as something else (it remains unclear what that would be), one would need to clearly define emotion contagion first, and then have some independent measures of that it occurs; e.g. increase in heart rate, matching facial expressions, etc. moreover, if the task was presented in a passive-viewing fashion without any specific instruction (see 2), subjects may have associated the different kinds of stimuli with each other in such an experimental context: Intense somatic pain will always (in healthy individuals) elicit a strong affective reaction, and on the other hand, intense facial expressions of pain are most likely associated with physical pain. To play devil's advocate, one could also label the two paradigms as "passive pain perception via facial expression" vs. "passive pain perception via somatosensory events"

We agree with the reviewer that it is necessary to conceptualize the mental process shared by the two stimuli sets in the context of the rationale of the present study. Based on the comments from the reviewers we included a more specific description of the shared mental process in the Introduction and in line with current conceptualizations of empathy which differentiate between contagion and empathy changed the wording to “vicarious sharing of pain”. We apologize for the unclear conceptual presentation in the initial version of the manuscript and included a clearer definition of the mental process in the Introduction. Moreover, we agree that referring to the stimuli sets as “physical” and “affective” pain is misleading. Based on the reviewers comment we changed the description of the two stimuli sets and included rationale for selecting these stimuli sets as follows: “Vicarious pain can be triggered by observing or imagining another individual’s painful state and can be elicited by multiple types of social cues, particularly the observation of an inflicted physical injury or a facial expression of pain (Decety and Ickes, 2009; Jauniaux et al., 2019; Vachon-Presseau et al., 2012; Yesudas and Lee, 2015). While stimulus depicting the noxious stimulation of body limbs [i.e., observation of noxious stimulation induced vicarious pain (NS vicarious pain)] provides objective cues about the sensory component of the observed pain, the observation of facial expressions of pain [i.e., facial expressions induced vicarious pain (FE vicarious pain)] is considered more subjective and indirect as the pain experience of the expresser need to be interpreted by the observer (Hadjistavropoulos et al., 2011; Vachon-Presseau et al., 2012)… Despite the different psychological domains engaged in the pain empathic response induced by NS and FE both elicit vicarious pain experience (Timmers et al., 2018), encompassing pain-specific processes such as recognizing and understanding the painful state of the other person and affective sharing of pain but also non-specific processes that are shared between pain and other non-painful experiences such as arousal and negative affect (Zaki et al., 2016)…”.

Finally, the reviewer mentions the “passive presentation” of the stimuli. More specifically the participants were asked to attentively watch the stimuli (we now also clarify this in the revised version). This approach is often referred to as “implicit” processing in contrast to “explicit” processing during which subjects are instructed to explicitly evaluate the painfulness of the presented stimuli. We decided to employ an implicit processing instruction because: (1) this allowed us to use similar instructions for the vicarious pain and negative emotional expression paradigm, and (2) our main focus was on pain empathic reactivity rather than more cognitive aspects of empathy. Previous studies using implicit / passive viewing paradigms including instructions to attentively or to passively view the stimuli reliably elicited pain empathic response [for acute pain infliction stimuli see e.g., (Meng et al., 2012; Yao et al., 2016); for painful faces see e.g. (Sheng and Han, 2012)] and a recent meta-analysis examined the effects of explicit instructions (e.g., to emphasize with the person or to explicitly indicate the level of experienced vicarious pain) versus implicit instructions on the neural correlates of pain empathy and, in line with the original studies, reported that both instructions increase neural activity in the core pain empathy networks (Timmers et al., 2018).

2) Insufficient motivation for focusing on mid-insulaa) A reviewer notes that "The authors base their mid-insula-restricted analysis ("Shared representations in the mid-insula") of shared representations on a) their own findings within the same manuscript and b) one paper reporting a specific role of the middle portion of the insula. Judging from the presented maps in Figure 3D, activity in the insula stretches from anterior to posterior portions. Also, e.g. Corradi-dell'Acqua et al. 2016 report a specific role of the anterior insula in vicarious pain. Authors need to justify this approach more clearly (or expand their ROI), especially because it could clearly affect the outcome of other analyses, such as the mid-insula classification approach for the IAPS paradigm."

We agree with the reviewer that the rationale for the specific focus on the mid-insula needs to be justified in detail. The main aim of the mid-insula focused analysis was to explore whether the shared neural representations in this region would be sufficient to predict vicarious pain. As pointed out by the reviewer the focus on the mid-insula was based on (1) our results suggesting convergent engagement of the bilateral mid-insula in vicarious pain processing and (2) previous literature suggesting a critical role of the mid-insula in pain-related processes including self-experienced as well as vicarious pain. Based on the comment from the reviewer we reformulated the specific aim of the mid-insula focused analyses clearer and included additional supporting information on the role of the mid-insula in pain and pain empathic processing as follows:

“…Across the analyses we observed overlapping activation and shared representations in the mid-insula (see also Figure 4—figure supplement 1 for convergent findings across searchlight sizes). Accumulating evidence suggest a critical role of the mid-insula in pain-related processes, including self-experienced as well as vicarious pain. In line with functional anatomical studies suggesting that the mid-insula receives nociceptive information from thalamic nuclei (Craig et al., 1994; Craig et al., 2000) intracerebral electrical stimulation of the mid-insula evokes pain sensations (Afif et al., 2010) and previous MVPA studies demonstrated distinct neural representations between pain and non-pain negative stimuli in the (right) mid-insula yet shared representations across self-experienced and vicarious pain (Corradi-Dell'Acqua et al., 2011), while a recent meta-analysis of conventional fMRI empathy studies reported that vicarious pain uniquely activates the bilateral mid-insula and MCC as compared to empathy for non-pain negative affective states (Timmers et al., 2018). Based on the specific role of the mid-insula in pain-related processes we further explored whether the mid-insula shared neural representations of NS and FE could be sufficient to predict vicarious pain”

The Corradi-Dell’Acqua et al., 2016, study reported a lateralized effect of AI, i.e., the right AI encoded vicarious pain and negative affect (disgust and unfair) in distinct patterns while the left AI exhibited shared representations for vicarious pain, self-pain and negative affect. Moreover, in another study (Corradi-Dell'Acqua et al., 2011) the authors showed shared neural representations between felt and seen pain as well as non-pain negative emotion with the bilateral AI ROI. Thus, we cannot make the conclusion that AI has a specific role in vicarious pain. Moreover, using whole-brain decoding approaches, which explains considerably more variance in predicting these processes as compared to local regions (which were used in Corradi-dell'Acqua et al. studies) (Kragel et al., 2018; Woo et al., 2017), we found that bilateral mid-insula, rather than AI, were reliable predictors to infer both NS and SE vicarious pain.

3) Additional Analyses? One of the reviewers wondered whether the authors have measured affect in the pain fMRI task besides emotional arousal, and whether the identified pattern would relate to how positive/negative the participants found the given pain stimuli. This might provide evidence that the results could be separated from general emotional arousal.

We thank the reviewer for this interesting idea which would allow us to better integrate the two paradigms and this could further strengthen our conclusion that the identified vicarious pain-predictive patterns are independent of general negative emotional arousal. However, the examination of associations between behavioral indices and neural decoders within (or across) the vicarious pain and negative emotional processing paradigm is limited due to (1) the use of a blocked design presentation and corresponding fMRI models on the individual level, and, (2) the use of an implicit processing paradigm during fMRI acquisition (subjects were instructed to attentively watch the stimuli, explicit ratings were acquired after the fMRI and rather served to additionally validate the properties of the stimuli). Given that multivariate patterns can be generalized to different paradigms and stimuli, for example, the NPS (Neurologic Pain Signature; Wager et al., 2013) responds to diverse types of evoked pain (heat, electrical, laser, mechanical and visceral) in N > 600 subjects across diverse population with an average effect size of d = 2.18 (Zunhammer et al., 2018), we employed a different strategy to determine the specificity of the vicarious pain predictive pattern. Briefly, we propose that if the vicarious pain decoders were actually tracking unspecific arousal experience the decoders should accurately predict negative vs. neutral stimuli in the emotional processing task although the paradigms were slightly different for a similar strategy of testing the specificity of a neural signature (see e.g., Chang et al., 2015). However, we found that vicarious pain decoders barely discriminated negative from neutral stimuli (although statistically significant the accuracies were less than 61%), suggesting that the identified patterns are not likely to relate to how negative/arousal the participants found the given pain stimuli. Moreover, we also developed a general negative experience decoder and found that it predicted vicarious pain less accurately than the cross-modality predictions (see below for details), further suggesting that the identified vicarious pain patterns capture information more than general negative experience.

4) Additional Analyses? The authors show that the patterns in pain task could not predict ratings in the emotional task. Is it possible to test whether that the reverse pattern is evident? (i.e. the pattern in the emotional task does not predict responses in the pain task.)

We thank the reviewer for this advice and agree that this analysis could reveal important additional information of the specificity of the neural representations and the association between negative processing and pain empathy. To address this question, we developed a negative emotion-predictive pattern based on the data from the IAPS paradigm. The pattern could accurately classify non-painful negative vs. neutral stimuli (accuracy = 86 ± 1.6% SE, *P* < 0.001, d = 2.07 with repeated 10-fold cross-validation procedures). We next applied the negative emotion-predictive pattern to the vicarious pain data and observed that this decoder could significantly predict NS vicarious pain vs. its control (cross-validated accuracy = 70 ± 3.0% SE, *P* < 0.001, d = 0.88) and FE vicarious pain vs. its control (cross-validated accuracy = 61 ± 3.2% SE, *P* < 0.001, d = 0.28). Given that vicarious pain stimuli were general more negative as compared to the control stimuli (and NS vicarious pain was more negative as compared with FE vicarious pain) these findings were expected.

Of note the accuracy and effect sizes were lower as compared to the between modality vicarious pain prediction for both, the prediction of FE vicarious pain signature on NS vicarious pain vs. its control (cross-validated accuracy = 78 ± 2.7% SE, *P* < 0.001, d = 1.00) and vice versa (cross-validated accuracy = 69 ± 3.0% SE, *P* < 0.001, d = 0.65), demonstrating that the vicarious pain decoders performed better for predicting cross-modality vicarious pain signatures as compared to the non-painful negative emotion decoder. Moreover, the mid-insula neural representation for the negative-predictive pattern could not significantly predict vicarious pain (accuracies = 40 ± 3.2% SE, 48 ± 3.2% SE for NS and FE vicarious pain respectively), suggesting that despite shared neural signatures between negative affective processing and vicarious pain processing in distributed neural networks the mid-insula plays a specific role in vicarious pain across modalities. Moreover, in contrast to the vicarious pain-predictive patterns neither the whole-brain or the mid-insula negative-predictive patterns could predict thermal pain intensity (whole-brain, r_196_ = 0.101, *P* = 0.157; mid-insula, r_196_ = -0.319), which additionally emphasizes the functional specificity of the pain-predictive pattern in the domain of pain-related processing. We included the new results in the revised version of the manuscript and suggest that these findings indicate that although non-painful negative emotional expressions share some neural representations with the representation of vicarious pain, the latter cannot be simply explained by the former and has a higher sensitivity and specificity to vicarious pain, in particular pain-associated processes.

“…using the emotional processing data, we developed a negative emotion-predictive pattern which could accurately classify non-painful negative vs. neutral stimuli (accuracy = 86 ± 1.6% SE, *p* < 0.001, d = 2.07 using repeated 10-fold cross-validation procedures). The negative emotion-predictive pattern could significant discriminate NS vicarious pain versus its control (cross-validated accuracy = 70 ± 3.0% SE, *P* < 0.001, d = 0.88) and FE vicarious pain versus its control (cross-validated accuracy = 61 ± 3.2% SE, *P* < 0.001, d = 0.28). Of note, accuracy and effect size are lower as compared to FE vicarious pain pattern’s prediction of NS vicarious pain (cross-validated accuracy = 78 ± 2.7% SE, *P* < 0.001, d = 1.00) and vice versa (cross-validated accuracy = 69 ± 3.0% SE, *P* < 0.001, d = 0.65) and the mid-insula negative-predictive pattern did not predict vicarious pain (accuracies = 40 ± 3.2% SE, 48 ± 3.2% SE for NS and FE vicarious pain, respectively). Moreover, in contrast to the pain-predictive patterns (see below for details) neither the whole-brain or the mid-insula negative-predictive pattern could predict thermal pain intensity (whole-brain, r_196_ = 0.101, *P* = 0.157; mid-insula, r_196_ = -0.319), which additionally emphasizes the functional specificity of the pain-predictive pattern in the domain of pain-related processing. Together these findings suggest that negative emotional processing might share some neural representations with vicarious pain, but that the whole-brain and mid-insula vicarious pain representations are more specific to the pain-related information.”

5) Framing/Discussion of Differencesa) Authors need to crisply articulate how the significance of observed differences in light of the many perceptual and psychological differences across tasks.

We agree with this comment and included a clearer description of the findings in the context of the task differences in Discussion.

“…In line with previous studies demonstrating that while NS vicarious pain provides objective cues about the sensory component of the observed pain the FE vicarious lacks such information and is more subjective and indirect as the pain experience of the expresser need to be interpreted by the observer (Hadjistavropoulos et al., 2011; Vachon-Presseau et al., 2012), the decreased accuracies and effect sizes in the cross-modality predictions additionally suggest partly distinguishable neural representations of NS and FE vicarious pain possibly reflecting the engagement of different component processes.”

b) For example, a reviewer notes that, "The matching between the two vicarious conditions, but also to the control or comparison conditions, is far from optimal, so that many differences as well as a lack of "sharedness"/common variance between paradigms could be simply explained by that. In one case, the "neutral" condition is not fully neutral, as the picture shows a potentially noxious object that could also be seen as "just having missed to inflict pain/injury"; in the other, a painful expression is compared to a fully neutral one, without a noxious object; thus (visual and affective) salience between conditions starkly differs, which makes it hard to compare them and to draw meaningful conclusions when doing so;"

We agree with the reviewer that the stimuli sets differ in several perceptual aspects, however, the main aim of the study was to determine shared representations between the stimuli sets rather than differences. Moreover, the cross-modality predictions employed two-alternative forced-choice tests, that is, we applied a signature (e.g., the NS vicarious pain pattern) to the cross-modality (e.g., FE stimuli) vicarious pain and control β maps using dot products (which generated one pattern expression per β map) and predicted the map with the higher pattern expression as vicarious pain condition for each subject separately. This procedure statistically equals applying the signature to the vicarious pain > control contrast (e.g., in this example, FE vicarious pain > FE control) for each subject and comparing the pattern expressions with 0 (higher than 0 means correct prediction). Given that within the same stimulus set the vicarious pain and control stimuli were well matched our procedure, to some extent, controlled for perceptual specific characteristics (e.g., general face processing).

Moreover, we agree with the reviewer that the painful conditions may differ with respect to additional characteristics such that the painful stimuli were additionally rated as more arousing and negative as compared to the control stimuli. However, the ratings clearly indicate that both sets of painful stimuli were rated as considerably more painful than the control stimuli which were generally rated as low painful (please see our response to 1b for details) and the lack of sensitivity of the vicarious pain patterns to differentiate high-arousal negative stimuli from neutral stimuli in the IAPS paradigm argues against the notion that unspecific differences may have contributed to the predictive models, specifically with respect to mid-insula expressions.

c) S/he notes that, "things get even more problematic when comparing data from the IAPS and the thermal pain conditions, which differ in many more aspects. In the IAPS, pictures of various emotionally evocative scenes are shown, thus some of the differences may be explained by higher intertrial variance and thus also lower "SNR" in the IAPS runs; moreover, salience might have differed as well – so any difference between conditions might be related to that, and the conclusion "it is not just emotional arousal that triggers the vicarious pain responses" is not conclusive (although I would be very much in favor of it); what could have been done to better match conditions would be e.g. to compare different facial emotion expressions matched for emotional arousal, so the "modality is not changed";

We agree with the reviewer that these two paradigms might differ in many aspects. However, multivariate patterns could be generalized to different paradigms and stimuli with common mental processes, for example, the NPS (Neurologic Pain Signature; Wager et al., 2013) responds to diverse types of evoked pain (heat, electrical, laser, mechanical and visceral) in N > 600 subjects across diverse population with an average effect size of d = 2.18 (Zunhammer et al., 2018). In support of this view, we used the IAPS data to develop a negative-predictive signature (please see response to comment #4 for more details) and applied this signature to negative emotional data from a previous study n = 182; (Chang et al., 2015), which employed a quite different paradigm (e.g., event design and trial by trail rating using a five-point Likert scale). We found that despite of the un-matched stimuli and paradigm our signature could predict high (average of rating 4 and 5) versus low (average of rating 1 and 2) negative stimuli with an accuracy of 92 ± 2% using a forced-choice test.

The logic of our approach is that a neural representation of vicarious pain should not simply reflect more general processes like negative affect or arousal (Zaki et al., 2016) and one way of testing this is by eliciting arousal and negative affect in other ways (e.g., non-painful high-arousal negative pictures) and testing the specificity of the vicarious pain signature. If our vicarious pain decoders were predicting arousal or negative affect they should also accurately predict non-painful negative stimuli from the IAPS data. Our results showed that the vicarious pain decoders could barely generalize to non-painful high-arousal negative stimuli, suggesting that the vicarious pain decoders are not sensitive to general processes like arousal and negative affect. Interesting, Chang et al., 2015, study included a few “painful” negative stimuli (e.g., heavily wounded people), and our general vicarious pain decoder predicted high versus low negative stimuli in their data with an accuracy of 67 ± 3.6%, which was higher than the accuracy (59 ± 2.1%) of predicting non-painful negative versus neutral stimuli used in the IAPS paradigm although the IAPS paradigm in our study was more similar to the vicarious pain paradigm (e.g., same sample and both employed block designs). Similar results were found with NS and FE vicarious pain decoders separately. These findings, to some extent, suggest that our vicarious pain decoders are more likely to predict “pain” specific information rather than general process like arousal or negative affect. However, we agree with the reviewer that between paradigm differences such as response amplitude and variance can influence decoding accuracies (e.g., Smith et al., 2011) and thus we cannot fully exclude that – in addition to the separable neural processes engaged in pain and emotional processing (Gilam et al., 2020) – may have contributed to the lack of a prediction of the negative emotion data with the vicarious pain signatures. Based on the suggestion from the reviewer we acknowledge this as limitations in the revised version as follows:

“Compared to the homogeneous stimuli within the conditions of the vicarious pain and the self-experienced pain paradigm the stimuli displaying emotional evocative scenes from the IAPS database may have led to a higher inter-trial variance in the negative processing experiment. Although the inter-stimulus variance should not systematically differ between the experimental conditions employed to develop the corresponding decoder we cannot fully exclude that this may have partly contributed to the low accuracies of the emotional processing decoder with respect to predicting self-experienced pain ratings.”

We fully agree with the reviewer that the conclusion "it is not just emotional arousal that triggers the vicarious pain responses" generally reaches far beyond the present findings. The inclusion of the IAPS data served to test whether the neural representation of vicarious pain – which due to the arousal differences between the vicarious pain and non-painful control stimuli may capture both mental processes – represents separable neural representations from general negative arousal. Together with the comparably low accuracies of the vicarious pain decoders to predict (non-painful) negative from neutral stimuli in the IAPS task the higher accuracies for NS vicarious pain to predict FE vicarious pain (and vice versa) suggest that the vicarious pain decoders share some overlapping neural representations with negative arousal yet that these are not able to fully capture the empathic pain responses. Based on the comment from the reviewer we emphasized this point in the revised Discussion as follows:

“…we demonstrated that these patterns were not sensitive to respond to the processing of non-painful high-arousal negative stimuli in the same sample, together with the findings showing that NS vicarious pain predicted FE vicarious pain (and vice versa) more accurately as compared with the predictions using a negative emotion decoder, suggesting that the common vicarious pain representations do not simply reflect shared unspecific processes of negative affect or arousal.”

d) And, "the same limitation (about limited matching between conditions) applies to the comparison to the thermal pain condition, which is a very different condition in terms of trial structure, "immediateness" of experience for the participants, and visual stimulation (apart from the sample size)

With respect to the IAPS paradigm we partly agree with the reviewer (see comment above), however, we do not follow that “the same limitation applies to the comparison to the thermal pain condition”. Of note the vicarious pain decoders robustly predicted levels of self-experienced pain suggesting a generalizability (rather than difference) across different modalities of “pain” induction (for a similar view on the strategy of generalization approaches across contexts and modalities to determine the specificity of neural decoders for specific mental processes, see also Kragel et al., 2018). With respect to the different sample sizes please see also response and new analysis / results summarized under point #6a.

6) Need for a sober and more complete discussion of implications, and perhaps an additional analysisa) A reviewer notes that "the vicarious pain pattern successfully distinguishes between different thermal pain levels, speaking for shared neural patterns between self-experienced pain and vicarious pain, which appears to be the most interesting (yet deemphasized) finding of the study. Yet, after repeatedly emphasizing the advantages of MVPA over mass-univariate techniques, the authors argue that "although the vicarious pain patterns could accurately predict the intensity of self-experienced pain this result does not imply that vicarious and somatic pain share common neural representations", without further elaboration. Dismissing this finding in such a way leaves the manuscript with the sole main finding that physical and affective vicarious pain show overlap also on a multivariate pattern level; thus what seems crucial is to carve out much better the discrepancy between the current results and the previous ones by Krishnan in eLife as well; as a reader, I am left wondering what is correct now – that there are, or there aren't "shared representations"? the interpretation "Krishnan et al. used different instructions" is of course one possible explanation, but what about sample size and power? Maybe Krishnan et al. was underpowered, and the shared patterns would have been found there as well? But then why should a passive observation paradigm be "better" in identifying shared patterns, if anything I would have expected the opposite as my own experience as well as prior work e.g. by Feng et al. (in NeuroImage) tell us that reduced attention if anything leads to less activation in insula, A/MCC, and the likes? (apart from the fact that a between sample prediction should also have less sensitivity). My proposal here would be to go much deeper into the discrepancy between the current and Krishnan's findings – that alone could be a very interesting paper, given the current sample size; one way to do that could be e.g. to "simulate" (e.g. by random drawing and permutation testing with N as in Krishnan) the lower sample size of Krishnan and to test whether or not the differences between studies remain."

We thank the reviewer for her/his helpful comments and suggestions. We agree with the reviewer that the sample size might play an important role. To this end we employed simulations with the NS vicarious pain and its control condition (which are similar to the stimuli used by Krishnan et al., 2016) with randomly select N=20, 40, 80, 120, 160 and 200 subjects (repeated 2,000 times). We found that with increasing sample size used to develop the decoder predictions (prediction-outcome correlation coefficient) of both,

Moreover, we found that when we randomly select 40 subjects – thus closely matching sample size and β images used in Krishnan et al., 2016, to train the predictive model we observed that 10.2% of the simulations failed to predict thermal pain levels and 20.1% of the simulations failed to predict subjective pain ratings (correlation coefficients were lower than 0.1395, corresponding to P = 0.05), which was, to some extent, consistent with results reported by Krishnan et al., 2016. However, as mentioned by the reviewer, other differences between the studies may additionally have contributed to the inconsistent findings, including different instructions or attention of the subjects. In contrast in Krishnan et al., 2016 paper the vicarious pain signature was developed based on the ratings of “how much pain they (subjects) might feel in the same situation as displayed in the picture”, which mainly captured more cognitive component of pain empathy, whereas in our study subjects have stronger engaged the affective pain empathy component while watching the vicarious pain stimuli implicitly (see above online rating results). As hypothesized by the reviewer the specific task instructions modulate the neural activation in pain empathy paradigms. A recent meta-analysis reported that – although both instructions engaged the core empathic networks – the mid-cingulate gyrus was more activated by cognitive/evaluative pain empathy paradigms, while the right inferior frontal gyrus and anterior insula were more activated by passive perceptual/affective pain empathy paradigms (Timmers et al., 2018).

Due to these differences a clear determination of the specific factors that may have led to the different findings between the studies is not possible, yet we agree with the reviewer that sample size may have been a contributing factor that should be discussed. We thank the reviewer for this comment and included the new analyses exploring the effects of sample size on the prediction in the reviewed manuscript and integrated the corresponding findings into the Discussion as follows:

“Moreover, we found that the present pattern could successfully predict pain experience during thermal heat stimulation while the VPS was not sensitive to self-experienced pain. The observed differences might be explained in terms of (1) the considerably larger sample size included in the present study and prediction accuracy (as reflected by prediction-outcome correlation) of self-experienced pain experience increased as a function of sample size used to develop the NS vicarious pain decoder (see additional analysis presented in Figure 7—figure supplement 2), and (2) differences between paradigms and instructions such that, for example, a recent meta-analysis of empathy for pain studies showed that the mid-cingulate gyrus was more activated by explicit cognitive/evaluative paradigms while the right inferior frontal gyrus and anterior insula were more activated by implicit perceptual/affective paradigms (Timmers et al., 2018).”

b) S/he notes that, "the discrepancies between the uni and multivariate analyses are another "gold mine" of this paper, but they are hardly addressed or explained. For example, the effect size (r) of the univariare analyses in Figure 1 is very low (0.171 -> ~4% common variance), whereas the MVPA analyses suggest much stronger, up to ~50%, correspondence. The authors need to seriously grapple with this in the Discussion.

We agree with the reviewer that the discrepancies between the univariate and multivariate analyses are very interesting and important. Please note that the correlation coefficient between NS and FE vicarious pain patterns (r=0.170, see **Figure 3A**) was similar to the univariate activation maps although the multivariate pattern expressions (i.e., repeated cross-validated NS and FE vicarious pain pattern expressions of NS and FE stimuli, respectively) were more similar (mean rs_474_ > 0.43). However, this finding does not necessarily imply similar correlations between local patterns and local activations (e.g., focusing on the mid-insula reveals also higher activation similarity in this study). Moreover, the current findings of the comparisons between univariate and multivariate analyses might not generalize to other stimuli or paradigms. We thank the reviewer for this excellent idea, however, given that in the current design corresponding conclusions would need to remain speculative we decided to not include an in-depth discussion on this point.

[Editors' note: further revisions were suggested prior to acceptance, as described below.]

The reviewers and I were enthusiastic about the revision:– The authors have taken on board all of my comments and presented detailed responses and further analyses of the data. All of these provide assurances that the results are robust. The new figures and supplementary figures are great and illustrate the data and the robustness of the findings well.– The authors have diligently responded to all my prior concerns and comments, and carried out the additional analyses I have suggested. I think the current version of this study/manuscript presents major advancement in the field both from theoretical and technical perspectives. I have no further comments.– The authors have been very responsive to the reviewer comments, and I would like to thank them for that. The paper has much improved as a consequence… Again, congratulations on a fine and huge piece of work, that certainly will advance the field.Nevertheless, they did have some suggestions for further enhancing the report."It would benefit the paper to a. delineate more clearly/accessibly, what the mid-insula entails (there is a reference to the GitHub and paper, which is already great, but not all readers will want to check), and how it "differs"/can be disambiguated from "anterior" and possibly also "posterior" IC (anterior has been often reported in empathy but also in salience research)"

We fully agree that a clearer description of the mid-insula mask that was used should be incorporated in the manuscript to facilitate transparency and comparison with previous research on this cytoarchitectonic and functional heterogenous region. We therefore incorporated the following information in the manuscript text and included a figure supplement to the corresponding figure (Figure 5):

“The mid-insula was defined based on the Human Connectome Project (HCP) multi-modal parcellation atlas (Glasser et al., 2016) (encompassing PoI2, FOP2, FOP3 and MI and available from the Cognitive and Affective Neuroscience Laboratory Github repository at https://github.com/canlab/Neuroimaging_Pattern_Masks; Figure 5—figure supplement 1 displays the mid-insula mask).”

I leave it to the authors to determine whether they wish to tackle the following additional analysis…"It might be useful to also test whether these other divisions (AI, PI) show similar or distinct effects as the mid-insula."Or, whether they would prefer to simply amend the manuscript to address this constructive suggestion:"Basically, it would be good to know whether the mid-insula cluster includes previously reported anterior insula clusters (at least partially), and whether what we should take away really is a stronger focus on mid rather than anterior insula in future empathy/vicarious pain studies and models."Again, I am comfortable with the authors eschewing additional analyses if they wish, but they need to address the substance of the reviewer's very reasonable suggestion prior to publication.

We fully agree with the reviewer and the editor that this reasonable and important suggestion should be addressed to incorporate our findings better into current models on the role of the insula in (pain) empathy. We hesitate to include additional analysis on the two other subregions of the insula because this would lead to a massive increase in the number of tests (i.e., four predictions were tested alone for the pain empathy paradigm, not including the additional predictions for the IAPS paradigm or the two pain experience measures in the independent dataset). We therefore decided to discuss the contribution of neural representations in the anterior versus mid-insula in the context of previous fMRI studies employing comparable multivariate methods (e.g., Corradi-Dell'Acqua et al., 2011; Corradi-Dell’Acqua et al., 2016). To determine the regional overlap of our findings with the previous reported MVPA results we mapped the peak voxels of the previous studies by Corradi-Dell’ Acqua and colleagues (Corradi-Dell'Acqua et al., 2011; Corradi-Dell’Acqua et al., 2016) on our results and found the previous reported peak voxels did neither overlap with our mid-insula mask nor the cluster exhibiting overlapping MVPA patterns on the whole-brain level. From our perspective particularly the latter is noteworthy given that it indicates that the inconsistent findings between the studies with respect to the identified insular region are not simply an artifact of the mid-insula mask. Of note, we did find overlapping clusters in the anterior and middle insula in univariate and searchlight-based decoding analyses. In contrast the whole-brain predictive models only converged on the middle but not the anterior insula. The whole-brain MVPA based predictive models are more sensitive and specific in predicting mental processes including (vicarious) pain and negative emotions (Chang et al., 2015; Kragel et al., 2018; Krishnan et al., 2016) and reliable predictive regions identified in the whole-brain MVPA are generally more “conservative” as compared with univariate activation and searchlight-based decoding analyses see Figure 2 and Figure 4 as well as previous studies (e.g., Chang et al., 2015). In line with our findings, a previous study employed the same whole-brain MVPA approach to predict NS vicarious pain induced by an evaluative paradigm also identified the bilateral mid-insula as reliable (q < 0.05, FDR corrected) predictive regions (Krishnan et al., 2016), further conforming that the mid-insula cortex is consistently predictive of vicarious pain across paradigms and samples. Nevertheless, based on the comments from the editor / reviewer we decided to further elaborate on the anterior versus middle insula in the context of the discussion of the Corradi-Dell’Acqua et al. findings as follows:

“Although overarching models of the neural basis and neuroimaging meta-analysis (Jauniaux et al., 2019; Timmers et al., 2018) emphasize the role of the anterior insula in pain empathy processing, accumulating evidence from studies examining shared and process-specific representations of vicarious pain suggest a specific role of the mid-insula in vicarious pain (Corradi-Dell'Acqua et al., 2011; Krishnan et al., 2016), whereas the (left) anterior insula also responded to negative stimuli in general (Corradi-Dell'Acqua et al., 2011) and across modalities (Corradi-Dell’Acqua et al., 2016). […] Together with the functional relevance of the mid-insula to predict objective and subjective pain experience in an independent sample and the contribution of this region to nociception as well as vicarious pain (Botvinick et al., 2005; Krishnan et al., 2016; Lamm et al., 2011; Timmers et al., 2018; Wager et al., 2013) our findings suggest that the shared representations in the mid-insula across vicarious pain induction procedures may specifically code the automatic pain sharing which resonates with embodies conceptualizations of vicarious pain (see e.g., Corradi-Dell'Acqua et al., 2011 for a convergent interpretation).”

**References**

<milestone-start id="_ENREF_2" /><milestone-end />Bird, G., and Viding, E. (2014). The self to other model of empathy: providing a new framework for understanding empathy impairments in psychopathy, autism, and alexithymia. Neuroscience and Biobehavioral Reviews, 47, 520-532.

<milestone-start id="_ENREF_9" /><milestone-end />De Vignemont, F., and Singer, T. (2006). The empathic brain: how, when and why? Trends Cogn Sci, 10(10), 435-441.

Decety, J. E., and Ickes, W. E. (2009). The social neuroscience of empathy. MIT Press.

Gilam, G., Gross, J. J., Wager, T. D., Keefe, F. J., and Mackey, S. C. (2020). What Is the Relationship between Pain and Emotion? Bridging Constructs and Communities. Neuron.

Jauniaux, J., Khatibi, A., Rainville, P., and Jackson, P. L. (2019). A meta-analysis of neuroimaging studies on pain empathy: investigating the role of visual information and observers’ perspective. Social cognitive and affective neuroscience, 14(8), 789-813.

Lamm, C., Rütgen, M., and Wagner, I. C. (2019). Imaging empathy and prosocial emotions. Neuroscience letters, 693, 49-53.

Smith, A. T., Kosillo, P., and Williams, A. L. (2011). The confounding effect of response amplitude on MVPA performance measures. Neuroimage, 56(2), 525-530.

Yao, S., Becker, B., Geng, Y., Zhao, Z., Xu, X., Zhao, W., Ren, P., and Kendrick, K. M. (2016). Voluntary control of anterior insula and its functional connections is feedback-independent and increases pain empathy. Neuroimage, 130, 230-240.

Zunhammer, M., Bingel, U., and Wager, T. D. (2018). Placebo effects on the neurologic pain signature: a meta-analysis of individual participant functional magnetic resonance imaging data. JAMA neurology, 75(11), 1321-1330.